# LOSSY COMPRESSION WITH DISTRIBUTION SHIFT AS ENTROPY CONSTRAINED OPTIMAL TRANSPORT

**Huan Liu**[1][*], **George Zhang**[2][*], **Jun Chen**[1], **Ashish Khisti**[2]
[1]McMaster University, [2]University of Toronto
{liuh127, chenjun}@mcmaster.ca
gq.zhang@mail.utoronto.ca, akhisti@ece.utoronto.ca

## ABSTRACT

We study an extension of lossy compression where the reconstruction distribution is different from the source distribution in order to account for distributional shift due to processing. We formulate this as a generalization of optimal transport with an entropy bottleneck to account for the rate constraint due to compression. We provide expressions for the tradeoff between compression rate and the achievable distortion with and without shared common randomness between the encoder and decoder. We study the examples of binary, uniform and Gaussian sources (in an asymptotic setting) in detail and demonstrate that shared randomness can strictly improve the tradeoff. For the case without common randomness and squared-Euclidean distortion, we show that the optimal solution partially decouples into the problem of optimal compression and transport and also characterize the penalty associated with fully decoupling them. We provide experimental results by training deep learning end-to-end compression systems for performing denoising on SVHN and super-resolution on MNIST suggesting consistency with our theoretical results.

## 1 INTRODUCTION

Using deep neural networks for lossy image compression has proven to be effective, with rate-distortion performance capable of dominating general-purpose image codecs like JPEG, WebP or BPG (Rippel & Bourdev, 2017; Agustsson et al., 2017; Mentzer et al., 2018). More recently, many of these works have included generative aspects within the compression to synthesize realistic elements when the rate is otherwise too low to represent fine-grained details (Tschannen et al., 2018; Agustsson et al., 2019; Mentzer et al., 2020). Though this has been found to deteriorate rate-distortion performance, it has generally resulted in more perceptually-pleasing image reconstruction by reducing artifacts such as pixelation and blur. Using a distributional constraint as a proxy for perceptual measure, several works have subsequently formalized this in a mathematical framework known as the rate-distortion-perception tradeoff (Blau & Michaeli, 2018; 2019; Matsumoto, 2018; 2019; Theis & Wagner, 2021; Yan et al., 2021; Zhang et al., 2021). As is conventional in lossy compression, these works address the scenario in which both low distortion, whereby each individual image reconstruction resembles the ground truth image, and closeness in distribution in which it is not easy to discriminate between image samples from the data-generating distribution and reconstruction distribution, are desirable.

The underlying ideal in conventional compression systems is to have perfect reconstruction with respect to some ground truth input. However this is not the case in applications such as denoising, deblurring, or super-resolution (SR), which require restoration from a degraded input image. In fact, in these cases a ground truth may not even be available. In such applications naturally the reconstruction distribution must match the original source rather than the degraded input distribution. A large body of literature has been devoted to various image restoration tasks, including several methods based on deep learning including both supervised (e.g., (Blau & Michaeli, 2018)) and unsupervised learning methods (e.g., (Wang et al., 2021b)). Although most of the literature exclusively treats compression

---

[*]Equal Contribution

and restoration separately, in many application they can co-occur. For example, the encoder which records a degraded image may not be co-located with the decoder, but must transmit a compressed version of the image over a digital network. In turn, the decoder must perform both decompression and restoration simultaneously.

To that end, we study an extension of lossy compression in which the reconstruction distribution is different than the source distribution to account for distributional shift due to processing. The problem can be described as a transformation from some source domain to a new target domain under a rate constraint, which generalizes optimal transport. This readily extends other works which view image restoration under the perception-distortion tradeoff (Blau & Michaeli, 2018) or under optimal transport (Wang et al., 2021b). It also provides a generalization of the rate-distortion-perception problem (Blau & Michaeli, 2019) where the reconstruction distribution must be close to the input distribution. Following (Theis & Agustsson, 2021; Theis & Wagner, 2021), we also utilize *common randomness* as a tool for compression in our setting. Our results are summarized as follows:

- We provide a formulation of lossy compression with distribution shift as a generalization of optimal transport with an entropy constraint and identify the tradeoff between the compression rate and minimum achievable distortion both with and without common randomness at the encoder and decoder. We identify conditions under which the structure of the optimal solution partially decouples the problems of compression and transport, and discuss their architectural implications. We study the examples of binary, uniform and Gaussian sources (in asymptotic regime) in detail and demonstrate the utility of our theoretical bounds.

- We train deep learning end-to-end compression systems for performing super-resolution on MNIST and denoising on SVHN. Our setup is *unsupervised* and to the best of our knowledge the first to integrate both compression and restoration at once using deep learning. We first demonstrate that by having common randomness at the encoder and decoder the achievable distortion-rate tradeoffs are lower than when such randomness is not present. Furthermore, we provide experimental validation of the architectural principle suggested by our theoretical analysis.

## 2 THEORETICAL FORMULATION

We consider a setting where an input $X \sim p_X$ is observed at the encoder, which is a degraded (e.g., noisy, lower resolution, etc) version of the original source. It must be restored to an output $Y \sim p_Y$ at the decoder, where $p_Y$ denotes the target distribution of interest. For example, if $X$ denotes a noise-corrupted image and $Y$ denotes the associated clear reconstruction, then $p_Y$ can be selected to match the distribution of the original source. We will assume $p_X$ and $p_Y$ are probability distributions over $\mathcal{X}, \mathcal{Y} \subseteq \mathbb{R}^n$ and require $X$ and $Y$ to be close with respect to some fidelity metric, which will be measured using a non-negative cost function $d(x, y)$ over $\mathcal{X} \times \mathcal{Y}$. We will refer to $d(\cdot, \cdot)$ as the distortion measure and assume that it satisfies $d(x, y) = 0$ if and only if $x = y$. We further assume that $X$ cannot be directly revealed to the decoder, but instead must be transmitted over a bit interface with an average rate constraint. Such a scenario occurs naturally in many practical systems when the encoder and decoder are not co-located such as communication systems or storage systems. As one potential application, when aerial photographs are produced for remote sensing purposes, blurs are introduced by atmospheric

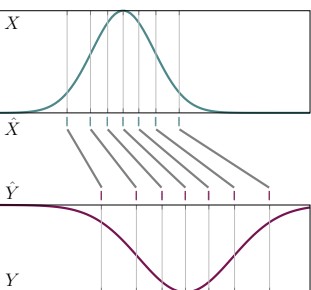

Figure 1: Illustration of Theorem 1 (no common randomness). Given source distribution $p_X$, target reconstruction distribution $p_Y$ and rate $R$, we can find quantizations $\hat{X}$ of $X$ and $\hat{Y}$ of $Y$ and consider transport between them.

turbulence, aberrations in the optical system and relative motion between camera and ground. In such scenarios unsupervised restoration is preferred as it is often intractable to accurately model such degradation processes and collection of paired training data can be time consuming or require significant human intervention. Unsupervised image restoration has been studied recently in Zhang et al. (2017); Pan et al. (2021); Wang et al. (2021b); Menon et al. (2020). These works also fix the reconstruction distribution $Y \sim p_Y$ and propose to minimize a distortion metric between the output and the degraded input as in our present work, but do not consider compression.

## 2.1 OPTIMAL TRANSPORT AND EXTENSIONS

**Definition 1** (Optimal Transport). Let $\Gamma(p_X, p_Y)$ be the set of all joint distributions $p_{X,Y}$ with marginals $p_X$ and $p_Y$. The classical optimal transport problem is defined as

$$D(p_X, p_Y) = \inf_{p_{X,Y} \in \Gamma(p_X, p_Y)} \mathbb{E}[d(X,Y)], \tag{1}$$

where we refer to each $p_{X,Y} \in \Gamma(p_X, p_Y)$ as a transport plan.

Operationally the optimal transport plan in (1) minimizes the average distortion between the input and output while keeping the output distribution fixed to $p_Y$. This may generate a transport plan with potentially unbounded entropy, which may not be amenable in a rate-constrained setting. We therefore suggest a generalization to Definition 1 which constrains the entropy of the transport plan. It turns out that having common randomness at the encoder and decoder can help in this setting, so we will distinguish between when it is available and unavailable.

**Definition 2** (Optimal Transport with Entropy Bottleneck — no Common Randomness). Let $M_{\mathrm{ncr}}(p_X, p_Y)$ denote the set of joint distributions $p_{X,Z,Y}$ compatible with the given marginal distributions $p_X, p_Y$ satisfying $p_{X,Z,Y} = p_X p_{Z|X} p_{Y|Z}$. The optimal transport from $p_X$ to $p_Y$ with an entropy bottleneck of $R$ and without common randomness is defined as

$$D_{\mathrm{ncr}}(p_X, p_Y, R) \triangleq \inf_{p_{X,Z,Y} \in M_{\mathrm{ncr}}(p_X, p_Y)} \mathbb{E}[d(X,Y)] \tag{2}$$
$$\text{s.t.} \quad H(Z) \leq R,$$

where $H(\cdot)$ denotes the Shannon entropy of a random variable.

We note that when the rate constraint $R$ is sufficiently large such that one can select $Z = X$ or $Z = Y$ in (2), then $D_{\mathrm{ncr}}(p_X, p_Y, R) = D(p_X, p_Y)$ in (1). More generally, $D(p_X, p_Y)$ serves as a lower bound for $D_{\mathrm{ncr}}(p_X, p_Y, R)$ for any $R > 0$. Definition 2 also has a natural operational interpretation in our setting. We can view the encoder as implementing the conditional distribution $p_{Z|X}$ to output a representation $Z$ given the input $X$, and the decoder as implementing the conditional distribution $p_{Y|Z}$ to output the reconstruction $Y$ given the representation $Z$. The entropy constraint $H(Z) \leq R$ essentially guarantees that the representation $Z$ can be losslessly transmitted at a rate close to $R$[1].

Of particular interest to us is the squared Euclidean distance $d(X,Y) = ||X - Y||^2$. As it turns out, when we specialize to this distance function we can without loss of optimality also impose a more structured architecture for implementing the encoder and the decoder. Let $W_2^2(\cdot, \cdot)$ be the squared quadratic Wasserstein distance, by using the squared loss in Definition 1.

**Theorem 1.** *Let*

$$D_{\mathrm{mse}}(p_X, p_Y, R) \triangleq \inf_{p_{\hat{X}|X}, p_{\hat{Y}|Y}} \mathbb{E}[||X - \hat{X}||^2] + \mathbb{E}[||Y - \hat{Y}||^2] + W_2^2(p_{\hat{X}}, p_{\hat{Y}}) \tag{3}$$
$$\text{s.t.} \quad \mathbb{E}[X|\hat{X}] = \hat{X}, \quad \mathbb{E}[Y|\hat{Y}] = \hat{Y}, \quad H(\hat{X}) \leq R, \quad H(\hat{Y}) \leq R,$$

*and*

$$D_{\mathrm{mse}}(p_X, R) \triangleq \inf_{p_{\hat{X}|X}} \mathbb{E}[||X - \hat{X}||^2] \tag{4}$$
$$\text{s.t.} \quad H(\hat{X}) \leq R.$$

*Moreover, let*

$$\overline{D}_{\mathrm{ncr}}(p_X, p_Y, R) \triangleq D_{\mathrm{mse}}(p_X, R) + D_{\mathrm{mse}}(p_Y, R) + W_2^2(p_{\hat{X}^*}, p_{\hat{Y}^*}), \tag{5}$$
$$\underline{D}_{\mathrm{ncr}}(p_X, p_Y, R) \triangleq D_{\mathrm{mse}}(p_X, R) + D_{\mathrm{mse}}(p_Y, R), \tag{6}$$

---

[1]The source coding theorem guarantees that any discrete random variable $Z$ can be losslessly compressed using a variable length code with average length of no more than $H(Z) + 1$ bits. We also note some differences between this formulation and the distortion-rate function from information theory - here the target distribution $p_Y$ is fixed, and we impose a constraint on entropy rather than mutual information to reflect that we perform compression on a per-sample basis (i.e. the one-shot scenario) rather than on asymptotically long blocks.

*where $p_{\hat{X}*}$ and $p_{\hat{Y}*}$ are the marginal distributions induced by the minimizers $p_{\hat{X}*|X}$ and $p_{\hat{Y}*|Y}$ that attain $D_{\mathrm{mse}}(p_X, R)$ and $D_{\mathrm{mse}}(p_Y, R)$, respectively (assuming the existence of such minimizers). Then under the squared Eucledian distortion measure,*

$$D_{\mathrm{ncr}}(p_X, p_Y, R) = D_{\mathrm{mse}}(p_X, p_Y, R). \tag{7}$$

*In addition, we have*

$$\overline{D}_{\mathrm{ncr}}(p_X, p_Y, R) \geq D_{\mathrm{ncr}}(p_X, p_Y, R) \geq \underline{D}_{\mathrm{ncr}}(p_X, p_Y, R), \tag{8}$$

*and both inequalities are tight when $p_X = p_Y$.*

Theorem 1 deconstructs $Z$ into the quantizations $\hat{X}$ of $X$ and $\hat{Y}$ of $Y$, and decomposes the overall distortion in (2) in terms of the losses due to quantization, transport, and dequantization in (3). It also suggests a natural architecture that partially decouples compression and transport without loss of optimality. First, the sender uses the distribution $p_{\hat{X}|X}$ to produce the compressed representation $\hat{X}$ from $X$. This is then passed through a "converter" $p_{\hat{Y}|\hat{X}}$ to transform $\hat{X}$ to an optimal representation $\hat{Y}$ of $Y$. Finally, the receiver maps $\hat{Y}$ back to $Y$ using the conditional distribution $p_{Y|\hat{Y}}$. This is illustrated in Figure 1. The entropy constraint $H(\hat{X}) \leq R$ in (2) essentially guarantees that $\hat{X}$ can be losslessly transmitted to the decoder where the converter can be applied to map $\hat{X}$ to $\hat{Y}$ before outputting $Y$. Alternately the constraint $H(\hat{Y}) \leq R$ guarantees that the converter could also be implemented at the encoder and then $\hat{Y}$ can be compressed and transmitted to the decoder. Finally note that our proposed architecture is symmetric[2] with respect to the encoder and the decoder and in particular the procedure to transport $Y$ to $X$ would simply be the inverse of transporting $X$ to $Y$, and indeed the distortion incurred by dequantizing $p_{Y|\hat{Y}}$ is the same as the distortion incurred by quantizing $p_{\hat{Y}|Y}$.

For the special case of same source and target distribution, we have $D_{\mathrm{mse}}(p_X, p_X, R) = 2D_{\mathrm{mse}}(p_X, R)$, implying that the rate required to achieve distortion $D$ under no output distribution constraint (and with the output alphabet relaxed to $\mathbb{R}^n$) achieves distortion $2D$ under the constraint that $Y$ equals $X$ in distribution. This recovers the result of Theorem 2 in Yan et al. (2021) for the one-shot setting. More generally, (8) shows that we may lower bound $D_{\mathrm{mse}}(p_X, p_Y, R)$ by the distortion incurred when compressing $X$ and $Y$ individually, each at rate $R$, through ignoring the cost of transport. On the other hand, the upper bound corresponds to choosing the optimal rate-distortion representations $\hat{X}^*, \hat{Y}^*$ for $X, Y$, then considering transport between them. The advantage of this approach is that knowledge of the other respective distribution is not necessary for design. Although not optimal in general, we will, in fact, provide an example where this is optimal in Section 2.2.

Finally, the following result implies that under mild regularity conditions, the optimal converter $p_{\hat{Y}|\hat{X}}$ can be realized as a (deterministic) bijection, and in the scalar case it can basically only take the form as illustrated in Figure 1.

**Theorem 2.** *Assume that $D_{\mathrm{ncr}}(p_X, p_Y, R)$ is a strictly decreasing function in a neighborhood of $R = R^*$ and $D_{\mathrm{ncr}}(p_X, p_Y, R^*)$ is attained by $p_{X,Z,Y}$. Let $\hat{X} \triangleq \mathbb{E}[X|Z]$ and $\hat{Y} \triangleq \mathbb{E}[Y|Z]$. Then*

$$H(\hat{X}) = H(\hat{Y}) = R^*, \tag{9}$$

$$\mathbb{E}[\|\hat{X} - \hat{Y}\|^2] = W_2^2(p_{\hat{X}}, p_{\hat{Y}}), \tag{10}$$

*and there is a bijection between $\hat{X}$ and $\hat{Y}$.*

We remark that in general computing the optimal transport map is not straightforward. For the case of binary sources we can compute an exact characterization for $D_{\mathrm{ncr}}$ as discussed in Section 2.2. Furthermore as discussed in Appendix A.6, $W_2^2(p_{\hat{X}}, p_{\hat{Y}})$ can be computed in closed form when $\hat{X}$ and $\hat{Y}$ are scalar valued, which can be used to obtain upper bounds on $D_{\mathrm{ncr}}$. In our experimental results in Section 3 we use deep learning based methods to learn approximately optimal mappings.

---

[2]We say that the problem is symmetric if it is invariant under reversing $p_X, p_Y$ with a new distortion measure defined by reversing the arguments of $d(\cdot, \cdot)$.

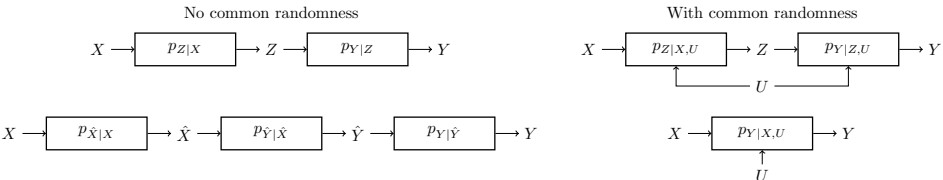

Figure 2: Architectures. Top left: Definition 2. Bottom left: Theorem 1. Top right: Definition 3. Bottom right: Theorem 3. Entropy coding of intermediate representations $Z, \hat{X}, \hat{Y}$ is not shown. For Theorem 3, the division between sender and receiver is across an encoder $C = f(X, U)$ and decoder $Y = g(C, U)$ performing entropy coding along $p_{Y|X,U}$.

So far we have focused on the setting when there is no shared common randomness between the encoder and the decoder. We will now consider the setting when a shared random variable denoted by $U$ is present at the encoder and decoder. We assume that the variable $U$ is independent of the input $X$ so that the decoder has no apriori information of the input. In practice the sender and receiver can agree on a pseudo-random number generator ahead of time and some kind of seed could be transmitted, after which both sides can generate the same $U$. We further discuss how shared randomness is used in practice in the experimental section.

**Definition 3** (Optimal Transport with Entropy Bottleneck — with Common Randomness). Let $M_{\mathrm{cr}}(p_X, p_Y)$ denote the set of joint distributions $p_{U,X,Z,Y}$ compatible with the given marginal distributions $p_X$, $p_Y$ and satisfying $p_{U,X,Z,Y} = p_U p_X p_{Z|X,U} p_{Y|Z,U}$, where $p_U$ represents the distribution of shared randomness. The optimal transport from $p_X$ to $p_Y$ with entropy bottleneck $R$ and common randomness is defined as

$$D_{\mathrm{cr}}(p_X, p_Y, R) \triangleq \inf_{p_{U,X,Z,Y} \in M_{\mathrm{cr}}(p_X, p_Y)} \mathbb{E}[d(X, Y)]$$
$$\text{s.t.} \quad H(Z|U) \leq R. \tag{11}$$

Note that we optimize over $p_U$ (the distribution associated with shared randomness), in addition to $p_{Z|X,U}$ and $p_{Y|Z,U}$ in (11). Furthermore, $D_{\mathrm{cr}}(p_X, p_Y, R) \leq D_{\mathrm{ncr}}(p_X, p_Y, R)$ in general, as we do not have access to shared randomness in Definition 2. Also from the same argument that was made following Definition 2, we have that $D_{\mathrm{cr}}(p_X, p_Y, R) \geq D(p_X, p_Y)$ in Definition 1. As with Definition 2, we can also provide a natural operational interpretation. In particular, given the input $X$ and common randomness $U$ the encoder can output a compressed representation $Z$ using the conditional distribution $p_{Z|X,U}$. The representation $Z$ can be losslessly compressed approximately to an average rate of $R$ again by exploiting the shared randomness $U$. Finally the decoder, given $Z$ and $U$ can output the reconstruction $Y$ using the conditional distribution $p_{Y|Z,U}$. An interesting difference with Definition 2 is that the setup is no longer symmetric between encoder and decoder, as $X$ is independent of $U$ but $Y$ is not. The following result provides a simplification to the architecture in Definition 3.

**Theorem 3.** Let $Q_{cr}(p_X, p_Y)$ denote the set of joint distributions $p_{U,X,Y}$ compatible with the given marginals $p_X$, $p_Y$ satisfying $p_{U,X,Y} = p_U p_X p_{Y|U,X}$ as well as $H(Y|U, X) = 0$. Then

$$D_{\mathrm{cr}}(p_X, p_Y, R) = \inf_{p_{U,X,Y} \in Q_{cr}(p_X, p_Y)} \mathbb{E}[d(X, Y)]$$
$$s.t. \quad H(Y|U) \leq R. \tag{12}$$

Before discussing the implications of Theorem 3 we remark on a technical point. Because the Shannon entropy is defined only for discrete random variables, $U$ must be chosen in a way such that $Y|U = u$ is discrete for each $u$, even for continuous $(X, Y)$. This is known to be possible, e.g., Li & El Gamal (2018) have provided a general construction for a $U$ with this property, with additional quantitative guarantees to ensure that $U$ is informative of $Y$. In the finite alphabet case we show in Appendix A.3 that optimization of $U$ can be formulated as a linear program.

We next discuss the implication of Theorem 3. First note that the problem can be modelled with only $p_{Y|U,X}$ producing a reconstruction $Y$ without the need for the intermediate representation $Z$, much like the conventional optimal transport in Definition 1. The condition $H(Y|U, X) = 0$ also implies that the transport plan is deterministic when conditioned on the shared randomness, which plays the role of stochasticity. Furthermore in this architecture the encoder should compute the representation

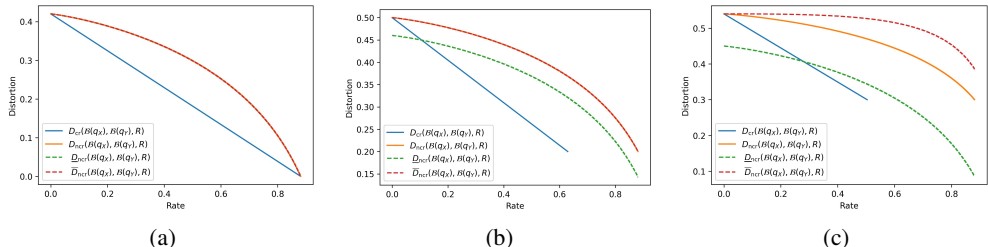

Figure 3: Binary case distortion-rate tradeoffs. (a) $q_X = q_Y = 0.3$, where $\overline{D}_{\mathrm{ncr}}(\mathcal{B}(q_X), \mathcal{B}(q_Y), R)$ and $\underline{D}_{\mathrm{ncr}}(\mathcal{B}(q_X), \mathcal{B}(q_Y), R)$ coincide with $D_{\mathrm{ncr}}(\mathcal{B}(q_X), \mathcal{B}(q_Y), R)$; (b) $q_X = 0.3, q_Y = 0.5$, where $\overline{D}_{\mathrm{ncr}}(\mathcal{B}(q_X), \mathcal{B}(q_Y), R)$ is tight but $\underline{D}_{\mathrm{ncr}}(\mathcal{B}(q_X), \mathcal{B}(q_Y), R)$ is loose; (c) $q_X = 0.3, q_Y = 0.6$, where both bounds are loose. Moreover, it can be seen from all these examples that common randomness can indeed help improve the distortion-rate tradeoff.

$Y$ given the source $X$ and the shared random-variable $U$ (which corresponds to the transport problem) and then compress it losslessly at a rate close to $H(Y|U)$ (which corresponds to the compression problem). The receiver only needs to decompress and reconstruct $Y$. This is in contrast to the case without common randomness in Theorem 1 where the $Y$ must be generated at the decoder.

## 2.2 Numerical Examples

We present how the results in Theorem 1 & 3 can be evaluated for some specific source models. We first consider the example of Binary sources. Let $X \sim \mathcal{B}(q_X)$ and $Y \sim \mathcal{B}(q_Y)$ be two Bernoulli random variables with $q_X, q_Y \in (0,1)$, and let $d(\cdot, \cdot)$ be the Hamming distortion measure $d_H(\cdot, \cdot)$ (i.e., $d_H(x, y) = 0$ if $x = y$ and $d_H(x, y) = 1$ otherwise), which coincides with the squared error distortion in Theorem 1 for binary variables. The explicit expressions of $D_{\mathrm{cr}}(\mathcal{B}(q_X), \mathcal{B}(q_Y), R)$, $D_{\mathrm{ncr}}(\mathcal{B}(q_X), \mathcal{B}(q_Y), R)$ as well as $\overline{D}_{\mathrm{ncr}}(\mathcal{B}(q_X), \mathcal{B}(q_Y), R)$ and $\underline{D}_{\mathrm{ncr}}(\mathcal{B}(q_X), \mathcal{B}(q_Y), R)$ are provided by Theorem 4 in Appendix A.4, from which the following observations can be made. In general, we have $D_{\mathrm{ncr}}(\mathcal{B}(q_X), \mathcal{B}(q_Y), R) > D_{\mathrm{cr}}(\mathcal{B}(q_X), \mathcal{B}(q_Y), R)$, i.e., common randomness strictly improves the distortion-rate tradeoff (except at some extreme point(s)). Moreover, as long as $\mathcal{B}(q_X)$ and $\mathcal{B}(q_Y)$ are biased toward the same symbol (namely, $q_X, q_Y \leq 1/2$ or $q_X, q_Y \geq 1/2$), the upper bound $\overline{D}_{\mathrm{ncr}}(\mathcal{B}(q_X), \mathcal{B}(q_Y), R)$ is tight, which implies that blindly using optimal quantizer and dequantizer in the conventional rate-distortion sense incurs no penalty. Some illustrative examples are shown in Figure 3.

In Appendix A.6 we consider the case when $X$ and $Y$ are continuous valued sources from a uniform distribution and establish an upper bound on $D_{\mathrm{ncr}}(\cdot)$ that is shown to be tight as the rate $R \to \infty$. For Gaussian distributions in the asymptotic optimal transport setting (see Appendix A.7 for relevant definitions and results) we present results qualitatively similar to the binary case in Appendix A.8.

## 3 Experimental Results

We use two sets of results to illustrate that the principles derived from our theoretical results are applicable to practical compression with deep learning. Importantly, we assume an *unsupervised* setting in which we have only unpaired noisy and clean images available to us, as in Wang et al. (2021b). Our first experiment is, to the best of our knowledge, the first in which restoration and compression are performed jointly using deep learning. We will furthermore demonstrate the utility of common randomness in this setting. The second set of experiments are designed on the principle of Theorem 1. In addition to the generator trained from our first experiment, we will construct a helper network to allow us to estimate the decomposition (3). This is then compared with the direct loss between the noisy image and rate-constrained denoising reconstruction. If the losses are close, this would suggest that the decomposition is not only without loss of optimality but also effective.

### 3.1 Rate-Distortion Comparison with Common Randomness

Let $p_X$ be a degraded source distribution that we wish to restore and $p_Y$ be the target distribution. Our goal is to compress $X$ so that the reconstruction semantically resembles $X$ within target distribution $p_Y$. For our application, we will use MSE loss as a fidelity criterion. Let $f$ be an encoder, $Q$ a quantizer, and $g$ a decoder. For a given rate $R$ with common randomness $U$ available, we have a

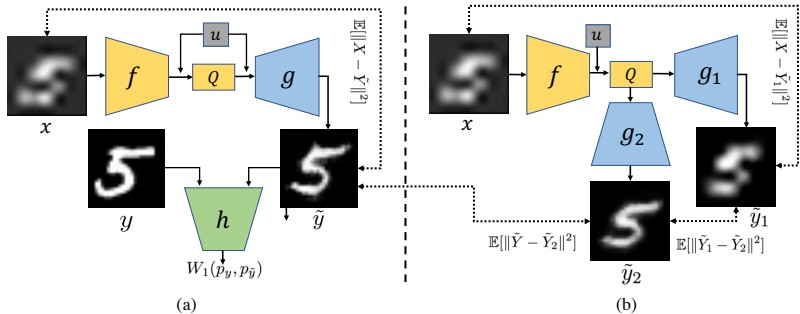

Figure 4: Illustration of our experimental setup. (a) shows the end-to-end learning system with common randomness, where the encoder and decoder have access to the same randomness $u$. (b) presents the network setup for verifying the architecture principle given in Theorem 1.

problem of the form

$$\min_{f,g,Q} \quad \|X - g(Q(f(X,U)))\|_2^2$$
$$\text{s.t.} \quad p_{g(Q(f(X,U)))} = p_Y, \quad H(Q(f(X,U))|U) \leq R, \tag{13}$$

which uses parameterized neural networks to implement (11). We also fix $Q$ such that a hard constraint on the rate is satisfied and assume $f$ and $g$ are sufficiently expressive to map to these fixed quantization points. Let $\tilde{Y} \triangleq g(Q(f(X,U)))$. We will use a penalty on the Wasserstein-1 distance between $p_{\tilde{Y}}$ and $p_Y$ in accordance with the Wasserstein GAN (Arjovsky et al., 2017) framework, so that our system is a stochastic rate-constrained autoencoder with GAN regularization. Specifically, we follow the network shown in Figure 4(a) which in addition to $f$, $Q$, and $g$ contains critic $h$.

For the realization of common randomness in Definition 3, we adopt the universal quantization scheme of Ziv (1985); Theis & Agustsson (2021). Given trained $f$ and $g$ and degraded image $X$, we generate restored image $\tilde{Y}$ through

$$\tilde{Y} = g(Q(f(X) + U) - U), \tag{14}$$

where $U$ is the stochastic noise shared by the sender and receiver. Details about the quantization are provided in Appendix B.2. To find an appropriate $f$ and $g$, we use the relaxed objective

$$L_1 = \mathbb{E}[\|X - \tilde{Y}\|^2] + \lambda W_1(p_Y, p_{\tilde{Y}}), \tag{15}$$

which is the sum of the MSE and Wasserstein-1 losses weighted by $\lambda$. By optimizing our network using this objective, we see two favorable properties. First, the Wasserstein-1 loss ensures the distribution of output is close to that of target images, i.e. $p_{\tilde{Y}} \approx p_Y$ for sufficiently large $\lambda$. Moreover, the MSE loss that pushes the output $\tilde{Y}$ to input $X$ ensures that the output structurally resembles $X$. Consequently, the training objective allows the output $\tilde{Y}$ to be clear and preserves content from input.

To generate a rate-distortion trade-off curve, we modify the encoder to produce a different number of symbols ranging from low bit rate to high bit rate and record the MSE distortion loss between noisy inputs and denoised outputs. Figure 5(a) and Figure 5(c) show the curves for image super-resolution and image denoising. We also show some qualitative results in Figure 5(b) and 5(d). As the rate increases, the generated high-quality images are clearer.

As exemplified by the numerical results in Section 2.2, common randomness can help reduce the rate that is needed for reconstruction given a specific distortion. Equivalently, given a fixed rate, a system with common randomness can perform better than one without common randomness. To demonstrate this in practice, we conduct the following experiment. We remove the common randomness setup from the framework in Section 3.1 and alternatively add two independent noises $U_1$ and $U_2$ to the encoder and decoder sides. Concretely, under the new setting, (14) becomes

$$\tilde{Y} = g(Q(f(X) + U_1) - U_2) \tag{16}$$

Then we conduct training using the objective (15) as in the common randomness. The tradeoff curve without common randomness for both tasks are shown in Figure 5(a) and 5(c) with orange dots. Performance of the framework is better when there is common randomness.

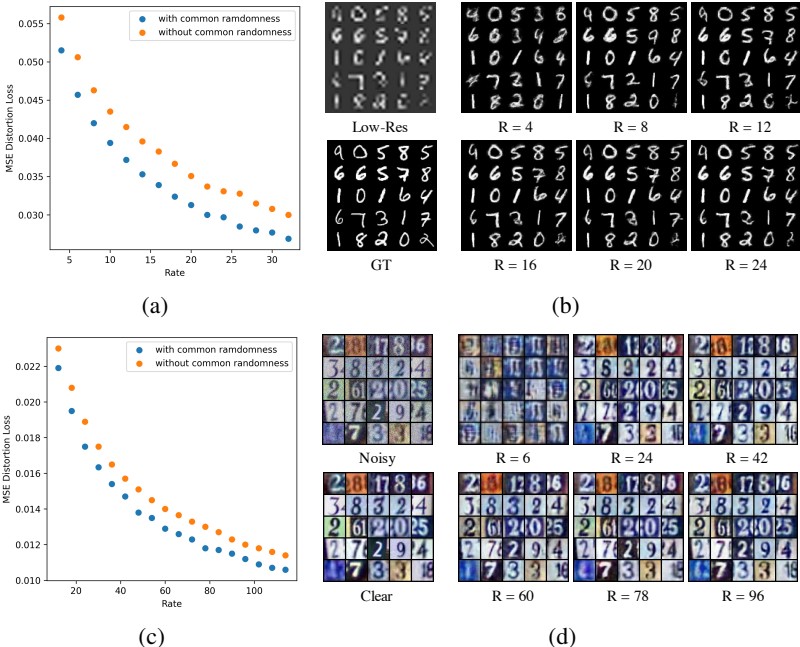

Figure 5: (a)(b) The experimental results of 4 times image super-resolution. (c)(d) The experimental results of image denoising. The noise pattern is synthesized by additive Gaussian noise with standard deviation set to 20. (a)(c) Rate-distortion trade-offs. Blue points are the MSE distortion loss for a particular rate under the setting of using common randomness, while orange points illustrate the same trade-off without using common randomness. For both tasks, at any rate, the performance of using common randomness is better than the case without common randomness. (b)(d) Examples for outputs from several models with different rates. As the rate increases, the outputs become clearer.

## 3.2 ARCHITECTURAL PRINCIPLE

In the case without common randomness, Theorem 1 implies that (under the rate constraint) the overall distortion $\mathbb{E}[\|X - Y\|^2]$ can be decomposed to the summation of the three distortion terms

$$\mathbb{E}[\|X - Y\|^2] = \mathbb{E}[\|X - \hat{X}\|^2] + \mathbb{E}[\|Y - \hat{Y}\|^2] + W_2^2(p_{\hat{X}}, p_{\hat{Y}}), \tag{17}$$

where $\hat{X}$ and $\hat{Y}$ are some representations of $X$ and $Y$ under MSE distortion. The chosen rate-distortion representations $\hat{X}$ for $X$ and $\hat{Y}$ for $Y$ must not only be representative of $X$ and $Y$, but also enjoy low cost of transport between one another. We now seek to estimate the overhead of this decomposition in practice.

However, due to the nature of the deep learning framework, the distortion measure between images and compressed representations cannot be explicitly measured. Thus, we alternatively develop a two-branch network to compare the summation of the three distortion components (17) to the overall distortion. First, we take trained $f$ and $g$ from the previous experiment and freeze their weights. Given noisy input $X$, we encode it through $f$, then decoder $g_1$ is trained to minimize the distortion with $X$, and decoder $g_2$ is trained to minimize the distortion with $\tilde{Y} = f(g(X))$ which distributionally approximates a clean restoration (we use $\tilde{Y}$ instead of ground truth because we assume an unsupervised setting). Let

$$\tilde{Y}_1 = g_1(f(X)), \quad \tilde{Y}_2 = g_2(f(X)).$$

The idea here is that $\tilde{Y}_1$ is a rate-constrained reconstruction of $X$ and $\tilde{Y}_2$ is a rate-constrained reconstruction of $\tilde{Y}$, both of which are produced from compressing $X$ using $f$. We assume this is reasonable as in light of Theorem 2, there is no loss of optimality in doing so given sufficiently expressive neural networks. The overall decomposed loss is then given by

$$L_2 = \underbrace{\mathbb{E}[\|X - \tilde{Y}_1\|^2]}_{(a)} + \underbrace{\mathbb{E}[\|\tilde{Y} - \tilde{Y}_2\|^2]}_{(b)} + \underbrace{\mathbb{E}[\|\tilde{Y}_1 - \tilde{Y}_2\|^2]}_{(c)}, \tag{18}$$

Table 1: Results of architecture principle in Theorem 1. The end-to-end loss and decomposed loss are very close for different rates.

| | Super-resolution | | | | Denoising | | | |
|---|---|---|---|---|---|---|---|---|
| Rate | 4 | 10 | 20 | 30 | 12 | 30 | 60 | 90 |
| End-to-end Loss | 0.0558 | 0.0435 | 0.0351 | 0.0308 | 0.0230 | 0.0175 | 0.0140 | 0.0123 |
| Decomposed Loss | 0.0586 | 0.0453 | 0.0349 | 0.0309 | 0.0243 | 0.0192 | 0.0158 | 0.0139 |

in which $\tilde{Y}_1$ approximates $\hat{X}$ and $\tilde{Y}_2$ approximates $\hat{Y}$. Training is performed jointly over $g_1$ and $g_2$.

One additional point is that $g_1$ and $g_2$ can also be trained separately, although in this case we can no longer assume that $f$ can be reused without loss of optimality, as this objective would not be equivalent to minimizing (18) (there is no control over (c)). We develop an additional experiment in which we optimize encoder-decoder pairs $f_1, g_1$ to minimize (a) and $\tilde{f}_2, g_2$ to minimize (b), where now $\tilde{Y}_1$ is produced using $f_1, g_1$ and $\tilde{Y}_2$ using $f_2, g_2$. In this setting, we aim to approximate the rate-distortion optimal $\hat{X}^*$ and $\hat{Y}^*$ corresponding to $\overline{D}_{\mathrm{ncr}}(p_X, p_Y, \cdot)$ from Theorem 1 using $\tilde{Y}_1$ and $\tilde{Y}_2$, and in doing this separate optimization it is clear that we will drive down (a) and (b) but increase (c). As it turns out, the resultant values (a) and (b) obtained during joint optimization are not much worse than the values from separate optimization. This provides evidence that in practice, the optimal rate-distortion representations (i.e. under objective (4)) can be leveraged for the general objective (18) without much loss of optimality, which further suggests that the encoder $f_1$ can be potentially trained without knowledge of $p_Y$ without much performance loss. These can be viewed in Table 2.

Table 2: Comparison of separate vs. joint training for (18). See the above paragraph for explanation.

| | Super-resolution | | | | | | | |
|---|---|---|---|---|---|---|---|---|
| Rate | 4 | | 10 | | 20 | | 30 | |
| Method | Joint | Separate | Joint | Separate | Joint | Separate | Joint | Separate |
| $\mathbb{E}[\|X - \tilde{Y}_1\|^2]$ | 0.0355 | 0.0356 | 0.0223 | 0.0214 | 0.0136 | 0.0113 | 0.0092 | 0.0083 |
| $\mathbb{E}[\|\tilde{Y} - \tilde{Y}_2\|^2]$ | 0.0227 | 0.0216 | 0.0222 | 0.0206 | 0.0191 | 0.0191 | 0.0172 | 0.0155 |
| | Denoising | | | | | | | |
| Rate | 12 | | 30 | | 60 | | 90 | |
| Method | Joint | Separate | Joint | Separate | Joint | Separate | Joint | Separate |
| $\mathbb{E}[\|X - \tilde{Y}_1\|^2]$ | 0.0191 | 0.0190 | 0.0146 | 0.0145 | 0.0117 | 0.0123 | 0.0104 | 0.0107 |
| $\mathbb{E}[\|\tilde{Y} - \tilde{Y}_2\|^2]$ | 0.0050 | 0.0046 | 0.0044 | 0.0040 | 0.0038 | 0.0035 | 0.0032 | 0.0030 |

## 4 RELATED WORKS

Sinkhorn distances (Cuturi, 2013) are a formulation of optimal transport with a penalty term corresponding to the mutual information between the source and target distributions. This has been studied in information theory literature (e.g. (Bai et al., 2020; Wang et al., 2021a)). For source coding in particular, Saldi et al. (2015a;b) consider common randomness with constrained output distribution. Blau & Michaeli (2018) evaluated a number of deep image restoration techniques and somewhat counter-intuitively demonstrated a tradeoff between optimizing for distortion and "perceptual quality", i.e. realism. Wang et al. (2021b) model the shift in distribution due to degradation as an optimal transport problem. However this work does not consider compression and their results are qualitatively different from ours. Meanwhile, output-constrained lossy compression has also been shown to improve perceptual quality (Tschannen et al., 2018), leading to the rate-distortion-perception framework (Blau & Michaeli, 2019). An analysis of the one-shot distortion-rate function was studied recently by (Elkayam & Feder, 2020). Our problem formulation is different as we assume the output distribution is fixed.

## 5 CONCLUSION AND FUTURE WORK

We consider the setting of lossy compression in which we compress across different source and target distributions. We formulate this as an entropy-constrained optimal transport problem and provide expressions for characterizing the tradeoff between compression rate and the minimum achievable distortion with and without shared common randomness. We also develop a number of architectural principles through our theoretical results and provide experimental validations by training deep learning models for super-resolution and denoising tasks over compressed representations. On the theory side it will be interesting to consider the case where there are either rate constraints on the amount of shared common randomness between the encoder and decoder or consider the case when the shared randomness is correlated with the source input, which can arise in many practical applications. On the practical side it will be interesting to experimentally study the a broader set of tasks under distribution shift where our theory could be applicable.

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

# A  THEORETICAL RESULTS

## A.1  DISTORTION-RATE VS RATE-DISTORTION FORMULATION

In addition to Definitions 2 and 3, we can equivalently define

$$R_{\mathrm{ncr}}(p_X, p_Y, D) \triangleq \inf_{p_{X,Z,Y} \in M_{\mathrm{ncr}}(p_X, p_Y)} H(Z)$$
$$\text{s.t.} \quad \mathbb{E}[d(X, Y)] \leq D,$$
(19)

$$R_{\mathrm{cr}}(p_X, p_Y, D) \triangleq \inf_{p_{U,X,Z,Y} \in M_{\mathrm{cr}}(p_X, p_Y)} H(Z|U)$$
$$\text{s.t.} \quad \mathbb{E}[d(X, Y)] \leq D.$$
(20)

$D_{\mathrm{ncr/cr}}(p_X, p_Y, R)$ and $R_{\mathrm{ncr/cr}}(p_X, p_Y, D)$ are monotonically decreasing in $R$ and $D$, respectively, so they are the inverse of each other. Sometimes it is more convenient to work with this rate-distortion formulation.

## A.2  PROOFS OF THEORETICAL RESULTS

*Proof of Theorem 1.* For any $p_{X,Z,Y} \in M_{\mathrm{ncr}}(p_X, p_Y)$ with $H(Z) \leq R$,

$$\mathbb{E}[\|X - Y\|^2] = \mathbb{E}[\|X - \mathbb{E}[X|Z]\|^2] + \mathbb{E}[\|Y - \mathbb{E}[Y|Z]\|^2] + \mathbb{E}[\|\mathbb{E}[X|Z] - \mathbb{E}[Y|Z]\|^2]$$
$$\geq D_{\mathrm{mse}}(p_X, p_Y, R),$$

where the last inequality follows from the definition of $D_{\mathrm{mse}}(p_X, p_Y, R)$ and the fact that

$$\max\{H(\mathbb{E}[X|Z]), H(\mathbb{E}[Y|Z])\} \leq H(Z) \leq R.$$

As a consequence, we must have $D_{\mathrm{ncr}}(p_X, p_Y, R) \geq D_{\mathrm{mse}}(p_X, p_Y, R)$. On the other hand, for any $p_{\hat{X}|X}, p_{\hat{Y}|Y}$ with $\mathbb{E}[X|\hat{X}] = \hat{X}$, $\mathbb{E}[Y|\hat{Y}] = \hat{Y}$, $H(\hat{X}) \leq R$, and $H(\hat{Y}) \leq R$, we can construct a joint distribution $p_{X,\hat{X},\hat{Y},Y}$ such that $X \leftrightarrow \hat{X} \leftrightarrow \hat{Y} \leftrightarrow Y$ form a Markov chain, $p_{X,\hat{X}} = p_X p_{\hat{X}|X}$, $p_{Y,\hat{Y}} = p_Y p_{\hat{Y}|Y}$, and $p_{\hat{X},\hat{Y}}$ satisfying $\mathbb{E}[\|\hat{X} - \hat{Y}\|^2] = W_2^2(p_{\hat{X}}, p_{\hat{Y}})$. Note that

$$\mathbb{E}[\|X - Y\|^2] = \mathbb{E}[\|X - \hat{X}\|^2] + \mathbb{E}[\|Y - \hat{Y}\|^2] + \mathbb{E}[\|\hat{X} - \hat{Y}\|^2]$$
$$= \mathbb{E}[\|X - \hat{X}\|^2] + \mathbb{E}[\|Y - \hat{Y}\|^2] + W_2^2(p_{\hat{X}}, p_{\hat{Y}}).$$
(21)

Let $Z \triangleq \hat{X}$. It can be verified that $p_{X,Z,Y} \in M_{\mathrm{ncr}}(p_X, p_Y)$ and $H(Z) = H(\hat{X}) \leq R$, which, together with (21), implies $D_{\mathrm{ncr}}(p_X, p_Y, R) \leq D_{\mathrm{mse}}(p_X, p_Y, R)$. This completes the proof of (7).

Dropping the term $W_2^2(p_{\hat{X}}, p_{\hat{Y}})$ in (3) yields

$$D_{\mathrm{ncr}}(p_X, p_Y, R) \geq \tilde{D}_{\mathrm{mse}}(p_X, R) + \tilde{D}_{\mathrm{mse}}(p_Y, R),$$

where

$$\tilde{D}_{\mathrm{mse}}(p_X, R) \triangleq \inf_{p_{\hat{X}|X}} \mathbb{E}[\|X - \hat{X}\|^2]$$
$$\text{s.t.} \quad \mathbb{E}[X|\hat{X}] = \hat{X}, \quad H(\hat{X}) \leq R.$$

and $\tilde{D}_{\mathrm{mse}}(p_Y, R)$ is definely analogously. On the other hand, choosing $p_{\hat{X}|X} = p_{\hat{X}'|X}$ and $p_{\hat{Y}|Y} = p_{\hat{Y}'|Y}$ in (3) gives

$$D_{\mathrm{ncr}}(p_X, p_Y, R) \leq \tilde{D}_{\mathrm{mse}}(p_X, R) + \tilde{D}_{\mathrm{mse}}(p_Y, R) + W_2^2(p_{\hat{X}'}, p_{\hat{Y}'}),$$

where $p_{\hat{X}'|X}$ and $p_{\hat{Y}'|Y}$ are the minimizers that attain $\tilde{D}_{\mathrm{mse}}(p_X, R)$ and $\tilde{D}_{\mathrm{mse}}(p_Y, R)$ respectively while $p_{\hat{X}'}$ and $p_{\hat{Y}'}$ are their induced marginal distributions. It is clear that $p_{\hat{X}'|X}$ and $p_{\hat{Y}'|Y}$ coincide

with $p_{\hat{X}^*|X}$ and $p_{\hat{Y}^*|Y}$ respectively as the constraints $\mathbb{E}[X|\hat{X}] = \hat{X}$ and $\mathbb{E}[Y|\hat{Y}] = \hat{Y}$ are automatically satisfied by $p_{\hat{X}^*|X}$ and $p_{\hat{Y}^*|Y}$. This proves (8). For the special case $p_X = p_Y$, we have $p_{\hat{X}^*|X} = p_{\hat{Y}^*|Y}$ and consequently the upper bound and the lower bound in (8) coincide.

Note that due to the involvement of conditional expectation, $\hat{X}$ is not necessarily defined over $\mathcal{X}$ if $\mathcal{X}$ is a strict subset of $\mathbb{R}^n$ (for the same reason, $\hat{Y}$ is not necessarily defined over $\mathcal{Y}$). In other words, the output of the quantizer is not consrained to the input alphabet and needs to be relaxed to $\mathbb{R}^n$. As such, $D_{\mathrm{mse}}(p_X, R)$ should be interpreted as the one-shot distortion-rate function with the reconstruction alphabet being $\mathbb{R}^n$, which is in general strictly below its counterpart with the reconstruction alphabet being $\mathcal{X}$ (also known as the distortion-rate-perception function with an inactive perception constraint) if $\mathcal{X}$ is a strictly subset of $\mathbb{R}^n$. This subtle issue, which is often overlooked in the literature, arises when one deals with discrete $X$ and $Y$ (see the binary example in Section 2.2 and Appendix A.4). $\square$

*Proof of Theorem 2.* We have that $\max\{H(\hat{X}), H(\hat{Y})\} \leq R^*$. If one of them, say $H(Y)$, is less than $R^*$, this will lead to a contradiction by the following argument. Note that

$$D^* = \mathbb{E}[\|X - Y\|^2] = \mathbb{E}[\|X - \hat{X}\|^2] + \mathbb{E}[\|Y - \hat{Y}\|^2] + \mathbb{E}[\|\hat{X} - \hat{Y}\|^2],$$

which depends on $p_{X,\hat{X},\hat{Y},Y}$ only through $p_{X,\hat{X}}$, $p_{\hat{X},\hat{Y}}$, and $p_{Y,\hat{Y}}$. We can construct a new joint distribution $p_{X,\hat{X}',\hat{Y}',Y}$ such that $p_{X,\hat{X}'} = p_{X,\hat{X}}$, $p_{\hat{X}',\hat{Y}'} = p_{\hat{X},\hat{Y}}$, $p_{Y,\hat{Y}'} = p_{Y,\hat{Y}}$, and $X \leftrightarrow \hat{X}' \leftrightarrow \hat{Y}' \leftrightarrow Y$ form a Markov chain. Denote $\hat{X}'$ by $Z'$. It is clear that the induced $p_{X,Z',Y}$ belongs to $M_{\mathrm{ncr}}(p_X, p_Y)$, preserves $\mathbb{E}[\|X - Y\|^2]$, and $H(Z') < R^*$, which is contradictory with the fact that $D_{\mathrm{ncr}}(p_X, p_{\hat{X}}, R)$ is a strictly decreasing function in a neighborhood of $R = R^*$ since $R$ can be set slightly below $R^*$ without violating the constraint $H(Z') \leq R$. This proves (9), which futher implies the existence of a bijection between $\hat{X}$ and $\hat{Y}$.

It remains to prove (10). If (10) does not hold, then we can find some $p_{\hat{X}'',\hat{Y}''}$ with $p_{\hat{X}''} = p_{\hat{X}}$ and $p_{\hat{Y}''} = p_{\hat{Y}}$ such that $\mathbb{E}[\|\hat{X}'' - \hat{Y}''\|^2] < \mathbb{E}[\|\hat{X} - \hat{Y}\|^2]$. Leverage this $p_{\hat{X}'',\hat{Y}''}$ to construct a new joint distribution $p_{X,\hat{X}'',\hat{Y}'',Y}$ such that $p_{X,\hat{X}''} = p_{X,\hat{X}}$, $p_{Y,\hat{Y}''} = p_{Y,\hat{Y}}$, and $X \leftrightarrow \hat{X}'' \leftrightarrow \hat{Y}'' \leftrightarrow Y$ form a Markov chain. Denote $\hat{X}''$ by $Z''$. It is clear that the induced $p_{X,Z'',Y}$ belongs to $M_{\mathrm{ncr}}(p_X, p_Y)$, $H(Z'') = R^*$, and

$$\begin{aligned}
\mathbb{E}[\|X - Y\|^2] &= \mathbb{E}[\|X - \hat{X}''\|^2] + \mathbb{E}[\|Y - \hat{Y}''\|^2] + \mathbb{E}[\|\hat{X}'' - \hat{Y}''\|^2] \\
&< \mathbb{E}[\|X - \hat{X}\|^2] + \mathbb{E}[\|Y - \hat{Y}\|^2] + \mathbb{E}[\|\hat{X} - \hat{Y}\|^2] \\
&= D^*,
\end{aligned}$$

which is contradictory with the fact that $R_{\mathrm{ncr}}(p_X, p_Y, D)$ is a strictly decreasing function in a neighborhood of $D = D^* \triangleq D_{\mathrm{ncr}}(p_X, p_{\hat{X}}, R^*)$ since $D$ can be set slightly below $D^*$ without violating the constraint $\mathbb{E}[\|X - Y\|^2] \leq D$. So in conclusion, the converter is a one-to-one mapping, which induces an optimal coupling that attains $W_2^2(p_{\hat{X}}, p_{\hat{Y}})$. $\square$

*Proof of Theorem 3.* Choosing $p_{U,X,Y}$ from $W_2^2(p_X, p_Y)$ and setting $Z = Y$ shows that

$$D_{\mathrm{cr}}(p_X, p_Y, R) \leq \inf_{p_{U,X,Y} \in W(p_X, p_Y)} \mathbb{E}[d(X, Y)]$$
$$\text{s.t.} \quad H(Y|U) \leq R.$$

So it remains to prove that this upper bound is tight. In the light of the functional representation lemma, for any $(U, X, Z, Y)$ with $p_{U,X,Z,Y} \in M_{\mathrm{cr}}(p_X, p_Y)$, there exist $V_1$, independent of $(U, X)$, and $V_2$, independent of $(U, X, V_1)$, as well as determintic mappings $\phi_1$ and $\phi_2$ such that $Z = \phi_2(U, X, V_1)$ and $Y = \phi_2(Z, V_2)$. Let $U' \triangleq (U, V_1, V_2)$. Clearly, $p_{U',X,Y} = p_{U'} p_X p_{Y|U',X}$ and $H(Y|U', X) = 0$. Moreover, we have

$$\begin{aligned}
H(Z|U) &\geq H(Z|U') \\
&\geq H(Y|U'),
\end{aligned}$$

where the last inequality is due to the fact that $Y$ is determined by $(Z, U')$. Therefore,

$$D_{\mathrm{cr}}(p_X, p_Y, R) \geq \inf_{p_{U',X,Y} \in W(p_X, p_Y)} \mathbb{E}[d(X, Y)]$$
$$\text{s.t.} \quad H(Y|U') \leq R.$$

This completes the proof of (12).

Note that each realization of $U$ is associated with a deterministic function from $\mathcal{X}$ to $\mathcal{Y}$. As a consequence, the problem boils down to optimizing the probablity distribution defined over this collection of functions. For the finite alphabet case, there are totally $|\mathcal{Y}|^{|\mathcal{X}|}$ such functions. In fact, a simple application of the support lemma shows that only $|\mathcal{Y}| + 1$ functions need to be assigned with a positive probability. $\qquad\square$

### A.3 LINEAR PROGRAM FORMULATION FOR COMMON RANDOMNESS

In the finite alphabet case, we can formulate Theorem 3 as follows:

$$D_{\mathrm{cr}}(p_X, p_Y, R) = \min_{p_U} \sum_{u \in \mathcal{U}} p_U(u)\mathbb{E}[d(X, f_u(X))]$$
$$\text{s.t.} \quad \sum_{u \in \mathcal{U}} p_U(u)H(f_u(X)) \leq R,$$
$$\sum_{u \in \mathcal{U}} p_U(u)\mathbb{P}(f_u(X) = y) = p_Y(y), \quad y \in \mathcal{Y},$$

where $p_U$ is defined over $\mathcal{U} \triangleq \{1, 2, \cdots, |\mathcal{Y}|^{|\mathcal{X}|}\}$, and $\{f_u : u \in \mathcal{U}\}$ is the set of all distinct functions from $\mathcal{X}$ to $\mathcal{Y}$. By the support lemma (Appendix C on page 631 of El Gamal & Kim (2011)), only $|\mathcal{Y}| + 1$ functions need to be assigned with a positive probability.

### A.4 BINARY CASE

Let $D_{\min}^{(B)} \triangleq |q_X - q_Y|$ and $D_{\max}^{(B)} \triangleq q_X + q_Y - 2q_X q_Y$. Note that $D_{\min}^{(B)}$ is the total variation distance between $\mathcal{B}(q_X)$ and $\mathcal{B}(q_Y)$, which is the minimum $\mathbb{E}[d_H(X, Y)]$ achievable by coupling $X$ and $Y$. On the other hand, we have $\mathbb{E}[d_H(X, Y)] = D_{\max}^{(B)}$ for $X, Y$ independent. It is clear that $D_{\min}^{(B)}$ and $D_{\max}^{(B)}$ are the infimum and supremum of $D_{\mathrm{ncr}}(\mathcal{B}(q_X), \mathcal{B}(q_Y), R)$ (as well as $D_{\mathrm{cr}}(\mathcal{B}(q_X), \mathcal{B}(q_Y), R)$), respectively.

**Theorem 4.** *Assume Hamming distortion measure. Under no common randomness, we have*

$$D_{\mathrm{ncr}}(\mathcal{B}(q_X), \mathcal{B}(q_Y), R) = \begin{cases} -\frac{2(1-q_X)(1-q_Y)}{1-H_b^{-1}(R)} + 2 - q_X - q_Y, & q_X + q_Y \leq 1, \\ -\frac{2q_X q_Y}{1-H_b^{-1}(R)} + q_X + q_Y, & q_X + q_Y > 1, \end{cases} \tag{22}$$

*for $R \in [0, \min\{H_b(q_X), H_b(q_Y)\}]$, and $= D_{\min}^{(B)}$ for $R > \min\{H_b(q_X), H_b(q_Y)\}$. Moreover,*

$$\overline{D}_{\mathrm{ncr}}(\mathcal{B}(q_X), \mathcal{B}(q_Y), R)$$
$$= \begin{cases} -\frac{2(1-q_X)(1-q_Y)}{1-H_b^{-1}(R)} + 2 - q_X - q_Y, & q_X, q_Y \leq \frac{1}{2}, \\ -\frac{2q_X q_Y}{1-H_b^{-1}(R)} + q_X + q_Y, & q_X, q_Y \geq \frac{1}{2}, \\ -\frac{(1-q_X)^2 + q_Y^2 - (q_Y - q_X + H_b^{-1}(R))^2}{1-H_b^{-1}(R)} + H_b^{-1}(R) - \frac{2q_Y(1-q_X)H_b^{-1}(R)}{(1-H_b^{-1}(R))^2}, & q_X < \frac{1}{2}, q_Y > \frac{1}{2}, \\ -\frac{q_X^2 + (1-q_Y)^2 - (q_X - q_Y + H_b^{-1}(R))^2}{1-H_b^{-1}(R)} + H_b^{-1}(R) - \frac{2q_X(1-q_Y)H_b^{-1}(R)}{(1-H_b^{-1}(R))^2}, & q_X > \frac{1}{2}, q_Y < \frac{1}{2}, \end{cases}$$
$$\tag{23}$$

*for $R \in [0, \min\{H_b(q_X), H_b(q_Y)\}]$, and*

$$\underline{D}_{\mathrm{ncr}}(\mathcal{B}(q_X), \mathcal{B}(q_Y), R) = \begin{cases} -\frac{(1-q_X)^2 + (1-q_Y)^2}{1 - H_b^{-1}(R)} + 2 - q_X - q_Y, & q_X, q_Y \le \frac{1}{2}, \\ -\frac{q_X^2 + q_Y^2}{1 - H_b^{-1}(R)} + q_X + q_Y, & q_X, q_Y \ge \frac{1}{2}, \\ -\frac{(1-q_X)^2 + q_Y^2}{1 - H_b^{-1}(R)} + 1 - q_X + q_Y, & q_X < \frac{1}{2}, q_Y > \frac{1}{2}, \\ -\frac{q_X^2 + (1-q_Y)^2}{1 - H_b^{-1}(R)} + 1 + q_X - q_Y, & q_X > \frac{1}{2}, q_Y < \frac{1}{2}, \end{cases} \tag{24}$$

*for $R \in [0, \min\{H_b(q_X), H_b(q_Y)\}]$. Here, $H_b^{-1}(R)$ denotes the inverse of the binary entropy function on $[0, 1/2]$. With common randomness,*

$$D_{\mathrm{cr}}(\mathcal{B}(q_X), \mathcal{B}(q_Y), R) = -\frac{2(1 - q_X)q_X R}{H_b(q_X)} + D_{\max}^{(B)} \tag{25}$$

*for $R \in [0, \rho H_b(q_X)]$, and $= D_{\min}^{(B)}$ for $R > \rho H_b(q_X)$. Here, $\rho \triangleq \min\{q_Y/q_X, (1-q_Y)/(1-q_X)\}$.*

*Proof of (25).* There are totally 4 distinct functions from $\{0, 1\}$ to $\{0, 1\}$:

$$f_1(x) = x, \quad f_2(x) = 1 - x, \quad f_3(x) = 0, \quad f_4(x) = 1, \quad x \in \{0, 1\}.$$

Therefore, we have

$$\sum_{u \in \mathcal{U}} p_U(u) H(f_u(X)) = H_b(q_X)(p_U(1) + p_U(2)),$$

$$\sum_{u \in \mathcal{U}} p_U(u) \mathbb{E}[d_H(X, f_u(X))] = p_U(2) + q_X p_U(3) + (1 - q_X)p_U(4),$$

$$\sum_{u \in \mathcal{U}} p_U(u) \mathbb{P}(f_u(X) = 1) = p_U(1)q_X + (1 - q_X)p_U(2) + p_U(4).$$

In light of Theorem 3,

$$R_{\mathrm{cr}}(\mathcal{B}(q_X), \mathcal{B}(q_Y), D) = \min_{p_U(1), \cdots, p_U(4)} H_b(q_X)(p_U(1) + p_U(2))$$

$$\text{s.t.} \quad p_U(2) + q_X p_U(3) + (1 - q_X)p_U(4) \le D, \tag{26}$$

$$q_X p_U(1) + (1 - q_X)p_U(2) + p_U(4) = q_Y, \tag{27}$$

$$p_U(1) + p_U(2) + p_U(3) + p_U(4) = 1, \tag{28}$$

$$p_U(1), p_U(2), p_U(3), p_U(4) \ge 0. \tag{29}$$

Note that

$$p_U(2) + q_X p_U(3) + (1 - q_X)p_U(4)$$

$$= p_U(2) + q_X p_U(3) + (1 - q_X)(1 - p_U(1) - p_U(2) - p_U(3)) \tag{30}$$

$$= -(1 - q_X)p_U(1) + q_X p_U(2) + (2q_X - 1)p_U(3) + 1 - q_X, \tag{31}$$

where (30) is due to (28). Moreover, it follows by (27) and (28) that

$$p_U(3) = -(1 - q_X)p_U(1) - q_X p_U(2) + 1 - q_Y. \tag{32}$$

Substituting (32) into (31) and invoking the fact that $p_U(2) \ge 0$ gives

$$p_U(2) + q_X p_U(3) + (1 - q_X)p_U(4)$$

$$= -2(1 - q_X)q_X(p_U(1) + p_U(2)) + 4(1 - q_X)q_X p_U(2) + D_{\max}^{(B)}$$

$$\ge -2(1 - q_X)q_X(p_U(1) + p_U(2)) + D_{\max}^{(B)},$$

which, together with (26), implies

$$p_U(1) + p_U(2) \ge \frac{1}{2(1 - q_X)q_X}(D_{\max}^{(B)} - D).$$

As a consequence, we must have

$$R_{\text{cr}}(\mathcal{B}(q_X), \mathcal{B}(q_Y), D) \geq \frac{H_b(q_X)}{2(1 - q_X)q_X}(D_{\max}^{(B)} - D).$$

One can readily verify that this lower bound is tight as it is attained by $p_U^*$ with

$$p_U^*(1) = \frac{1}{2(1 - q_X)q_X}(D_{\max}^{(B)} - D),$$

$$p_U^*(2) = 0,$$

$$p_U^*(3) = -\frac{1}{2q_X}(D_{\max}^{(B)} - D) + 1 - q_Y,$$

$$p_U^*(4) = -\frac{1}{2(1 - q_X)}(D_{\max}^{(B)} - D) + q_Y,$$

which satisfies (26)–(29) for $D \in [D_{\min}^{(B)}, D_{\max}^{(B)}]$. The expression of $D_{\text{cr}}(\mathcal{B}(q_X), \mathcal{B}(q_Y), R)$ can be obtained by taking the inverse of $R_{\text{cr}}(\mathcal{B}(q_X), \mathcal{B}(q_Y), D)$. $\qquad\square$

*Proof of (22).* We will rely on some results which will come after this proof.

Note that Hamming distortion coincides with squared error distortion when $\mathcal{X} = \mathcal{Y} = \{0, 1\}$. So Theorem 1, Lemma 1, and Lemma 2 are applicable here. In particular, in light of Lemmas 1 and 2, for any $R \geq 0$ and $\epsilon > 0$, there exists a joint distribution $p_{X\hat{X}\hat{Y}Y}$ compatible with the given marginal distributions $p_X$ and $p_Y$ such that $\hat{X}$ and $\hat{Y}$ are deterministically related finite-support random variables with $H(\hat{X}) = H(\hat{Y}) \leq R$ and $X \leftrightarrow \hat{X} \leftrightarrow \hat{Y} \leftrightarrow Y$ form a Markov chain; moreover, $\hat{X} = \mathbb{E}[X|\hat{X}]$, $\hat{Y} = \mathbb{E}[Y|\hat{Y}]$, and

$$\mathbb{E}[\|X - \hat{X}\|^2] + \mathbb{E}[\|Y - \hat{Y}\|] + \mathbb{E}[\|\hat{X} - \hat{Y}\|^2] \leq D_{\text{ncr}}(\mathcal{B}(q_X), \mathcal{B}(q_Y), R) + \epsilon. \qquad (33)$$

Without loss of generality, we assume $\hat{X}$ and $\hat{Y}$ take value from $\{\hat{x}_i\}_{i=1}^N$ and $\{\hat{y}_i\}_{i=1}^N$, respectively, and $\hat{Y} = \psi(\hat{X})$, where $\psi$ is a bijection from $\{\hat{x}_i\}_{i=1}^N$ to $\{\hat{y}_i\}_{i=1}^N$ with $\hat{y}_i = \psi(\hat{x}_i)$, $i = 1, \cdots, N$. Let $\theta_i \triangleq p_{\hat{X}}(\hat{x}_i)$, or equivalently, $\theta_i \triangleq p_{\hat{Y}}(\hat{y}_i)$, $i = 1, \cdots, N$. Note that $\hat{X} = \mathbb{E}[X|\hat{X}]$ and $\hat{Y} = \mathbb{E}[Y|\hat{Y}]$ if and only if $p_{X|\hat{X}}(1|\hat{x}_i) = \hat{x}_i$ and $p_{Y|\hat{Y}}(1|\hat{y}_i) = \hat{y}_i$ for $\theta_i > 0$, $i = 1, \cdots, N$. So the constraints $\sum_{i=1}^N p_{\hat{X}}(\hat{x}_i)p_{X|\hat{X}}(1|\hat{x}_i) = q_X$ and $\sum_{i=1}^N p_{\hat{Y}}(\hat{y}_i)p_{Y|\hat{Y}}(1|\hat{y}_i) = q_Y$ can be written equivalently as $\sum_{i=1}^N \theta_i\hat{x}_i = q_X$ and $\sum_{i=1}^N \theta_i\hat{y}_i = q_Y$. Moreover, it is easy to verify that

$$\mathbb{E}[\|X - \hat{X}\|^2] + \mathbb{E}[\|Y - \hat{Y}\|^2] + \mathbb{E}[\|\hat{X} - \hat{Y}\|^2]$$

$$= \sum_{i=1}^N \theta_i(1 - \hat{x}_i)\hat{x}_i + \sum_{i=1}^N \theta_i(1 - \hat{y}_i)\hat{y}_i + \sum_{i=1}^n \theta_i(\hat{x}_i - \hat{y}_i)^2$$

$$= \sum_{i=1}^N \theta_i(\hat{x}_i + \hat{y}_i - 2\hat{x}_i\hat{y}_i).$$

In light of Theorem 1 and (33), the following optimization problem (P) yields an upper bound on $D_{\text{ncr}}(p_X, p_Y, R)$ with a gap at most $\epsilon$:

$$\min_{(\theta_i, \hat{x}_i, \hat{y}_i)_{i=1}^N} \sum_{i=1}^N \theta_i(\hat{x}_i + \hat{y}_i - 2\hat{x}_i\hat{y}_i) \qquad (\text{P})$$

$$\text{s.t.} \quad \sum_{i=1}^N \theta_i \log\frac{1}{\theta_i} \leq R, \quad \sum_{i=1}^N \theta_i = 1, \quad \sum_{i=1}^N \theta_i\hat{x}_i = q_X, \quad \sum_{i=1}^N \theta_i\hat{y}_i = q_Y,$$

$$\theta_i \geq 0, \quad \hat{x}_i \in [0, 1], \quad \hat{y}_i \in [0, 1], \quad i = 1, \cdots, N.$$

Given $(\theta_i, \hat{y}_i)_i^N$, (P) degenerates to a linear programming problem with respect $(\hat{x}_i)_{i=1}^N$ over hyper-rectangle $[0, 1]^N$ subject to the constraint $\sum_{i=1}^N \theta_i\hat{x}_i = q_X$, for which the minimum is attained at

a point on an edge of $[0,1]^N$. Therefore, it suffices to consider $(\hat{x}_i)_{i=1}^N$ with at most one element different from 0 and 1. By a similar argument, it can be shown that there is no loss of optimality in assuming that at most one of $\hat{y}_i$, $i = 1, \cdots, N$, takes value other than 0 or 1. Due to the merge of different elements, the one-to-one relationship might not be preserved. Nevertheless, by Lemma 2, we just need to consider deterministically related $\hat{X}$ and $\hat{Y}$ with support size at most 3. Applying the linear programming argument to (P) with $N = 3$ shows that, at the cost of potentially compromising the one-to-one relationship, at most one element in the support of $\hat{X}$ as well as the support of $\hat{Y}$ need to take value different from 0 and 1. In the case that the bijection is lost, $\hat{X}$ or $\hat{Y}$ must have a reduced support size. One can restore the bijection by invoking Lemma 2, then use the linear programming argument to assign extreme values to all but at most one element in the support. Following this line of reasoning, we can conclude that the attention can be restricted to deterministically related $\hat{X}$ and $\hat{Y}$ with support size at most 3 and at most one element in the support different from 0 and 1. Moreover, the following configurations can be excluded.

1. Support size $= 3$ and the existence of pairs $(\hat{x}, \hat{y})$ and $(\hat{x}', \hat{y}')$ for some $\hat{x} > \hat{x}'$ and $\hat{y} < \hat{y}'$ ($(\hat{x}, \hat{y})$ is said to be a pair if $\hat{X} = \hat{x} \Leftrightarrow \hat{Y} = \hat{y}$): Since

$$
(\hat{x} - \hat{y})^2 + (\hat{x}' - \hat{y}')^2 - (\hat{x} - \hat{y}')^2 - (\hat{x}' - \hat{y})^2
$$
$$
= -2\hat{x}\hat{y} - 2\hat{x}'\hat{y}' + 2\hat{x}\hat{y}' + 2\hat{x}'\hat{y}
$$
$$
= 2(\hat{x} - \hat{x}')(\hat{y}' - \hat{y})
$$
$$
> 0,
$$

it follows that $\mathbb{E}[\|\hat{X} - \hat{Y}\|^2]$ can be strictly reduced by moving the same amount of probability from $\{\hat{X} = \hat{x}, \hat{Y} = \hat{y}\}$ to $\{\hat{X} = \hat{x}, \hat{Y} = \hat{y}'\}$ and from $\{\hat{X} = \hat{x}', \hat{Y} = \hat{y}'\}$ to $\{\hat{X} = \hat{x}', \hat{Y} = \hat{y}\}$. This modification does not affect $p_{\hat{X}}$ and $p_{\hat{Y}}$, and consequently $H(\hat{X})$, $H(\hat{Y})$, $\mathbb{E}[\|X - \hat{X}\|^2]$, $\mathbb{E}[\|Y - \hat{Y}\|^2]$ remain the same. So the distortion-rate performance of this configuration is strictly suboptimal.

2. Support size $= 2$ and the existence of pairs $(\hat{x}, \hat{y})$ and $(\hat{x}', \hat{y}')$ for some $\hat{x} > \hat{x}'$ and $\hat{y} < \hat{y}'$: Same as configuration 1).

3. Support size $= 2$ and existence of pairs $(\hat{x}, 1)$ and $(0, \hat{y})$ for some $\hat{x} \in (0, 1)$ and $\hat{y} \in (0, 1)$: It follows by $\mathbb{E}[X|\hat{X}] = \hat{X}$ and $\mathbb{E}[Y|\hat{Y}] = \hat{Y}$ that

$$
p_{\hat{X}, \hat{Y}}(\hat{x}, 1) = \frac{q_X}{\hat{x}},
$$
$$
p_{\hat{X}, \hat{Y}}(0, \hat{y}) = 1 - \frac{q_X}{\hat{x}},
$$
$$
\hat{y} = 1 - \frac{1 - q_Y}{1 - \frac{q_X}{\hat{x}}}.
$$

Clearly, $H(\hat{X}) = H(\hat{Y}) = H_b(\frac{q_X}{\hat{x}})$. Since $\hat{x} \in (0,1)$ and $\hat{y} \in (0,1)$, we must have $q_X < \frac{q_X}{\hat{x}} < q_Y$, which implies $H(\hat{X}) = H(\hat{Y}) > \min\{H(X), H(Y)\}$. Furthermore, it can be verified that

$$
\mathbb{E}[\|X - \hat{X}\|^2] + \mathbb{E}[\|Y - \hat{Y}\|^2] + \mathbb{E}[\|\hat{X} - \hat{Y}\|^2]
$$
$$
= \frac{q_X}{\hat{x}}\hat{x}(1 - \hat{x}) + \left(1 - \frac{q_X}{\hat{x}}\right)\hat{y}(1 - \hat{y}) + \frac{q_X}{\hat{x}}(1 - \hat{x})^2 + \left(1 - \frac{q_X}{\hat{x}}\right)\hat{y}^2
$$
$$
= q_Y - q_X.
$$

However, this end-to-end distortion is obviously achievable when $R = \min\{H(X), H(Y)\}$. So the rate-distortion performance of this configuration is strictly suboptimal.

4. Support size $= 2$ and the existence of pairs $(\hat{x}, 0)$ and $(1, \hat{y})$ for some $\hat{x} \in (0, 1)$ and $\hat{y} \in (0, 1)$: Same as configuration 3).

In view of the excluded configurations, we are left with the case where $p_{\hat{X}\hat{Y}}$ assigns all probabilities to $\{\hat{X} = 0, \hat{Y} = 0\}$, $\{\hat{X} = 1, \hat{Y} = 1\}$, and $\{\hat{X} = \hat{x}, \hat{Y} = \hat{y}\}$ for some $\hat{x} \in [0, 1]$ and $\hat{y} \in [0, 1]$. So it suffices to consider the $N = 3$ version of (P) with $\hat{x}_1 = \hat{y}_1 = 0$, $\hat{x}_3 = \hat{y}_3 = 1$, $\hat{x}_2 = \hat{x}$, and $\hat{y}_2 = \hat{y}$. The constraints $\sum_{i=1}^3 \theta_i = 1$, $\sum_{i=1}^3 \theta_i \hat{x}_i = q_X$, and $\sum_{i=1}^3 \theta_i \hat{y}_i = q_Y$ imply

$$\theta_1 = 1 - q_X - (1 - \hat{x})\theta,$$
$$\theta_3 = q_X - \hat{x}\theta,$$
$$\hat{y} = \frac{q_Y - q_X}{\theta} + \hat{x}.$$

In this way, we get a simplified optimization problem (P'):

$$\min_{\theta, \hat{x}} \ 2\hat{x}(1 - \hat{x})\theta + (1 - 2\hat{x})(q_Y - q_X) \qquad \text{(P')}$$

s.t. $\quad (1 - q_X - (1 - \hat{x})\theta) \log \dfrac{1}{1 - q_X - (1 - \hat{x})\theta} + \theta \log \dfrac{1}{\theta} + (q_X - \hat{x}\theta) \log \dfrac{1}{q_X - \hat{x}\theta} \leq R,$

$\quad \hat{x} \in [0, 1], \quad \theta \in [0, 1], \quad (1 - \hat{x})\theta \in [q_Y - q_X, 1 - q_X], \quad \hat{x}\theta \in [q_X - q_Y, q_X].$

Note that (P') does not depend on $\epsilon$ and consequently yields the exact characterization of $D_{\mathrm{ncr}}(\mathcal{B}(q_X), \mathcal{B}(q_Y), R)$. Therefore, $R_{\mathrm{ncr}}(\mathcal{B}(q_X), \mathcal{B}(q_Y), D)$ is characterized by the following optimization problem (P''):

$$\min_{\theta, \hat{x}} \ (1 - q_X - (1 - \hat{x})\theta) \log \frac{1}{1 - q_X - (1 - \hat{x})\theta} + \theta \log \frac{1}{\theta} + (q_X - \hat{x}\theta) \log \frac{1}{q_X - \hat{x}\theta} \qquad \text{(P'')}$$

s.t. $\quad 2\hat{x}(1 - \hat{x})\theta + (1 - 2\hat{x})(q_Y - q_X) \leq D,$
$\quad \hat{x} \in [0, 1], \quad \theta \in [0, 1], \quad (1 - \hat{x})\theta \in [q_Y - q_X, 1 - q_X], \quad \hat{x}\theta \in [q_X - q_Y, q_X].$

Given $\hat{x}$, the objective function of (P'') is concave in $\theta$ and consequently its minimum is attained at an endpoint of $[\underline{\theta}, \overline{\theta}]$, where

$$\underline{\theta} \triangleq \max \left\{ 0, \frac{q_Y - q_X}{1 - \hat{x}}, \frac{q_X - q_Y}{\hat{x}} \right\},$$
$$\overline{\theta} \triangleq \min \left\{ 1, \frac{1 - q_X}{1 - \hat{x}}, \frac{q_X}{\hat{x}}, \frac{D - (1 - 2\hat{x})(q_Y - q_X)}{2\hat{x}(1 - \hat{x})} \right\}.$$

Without loss of generality, we assume $q_Y \geq q_X$ and $q_X + q_Y \leq 1$. The following statements can be easily verified.

1. For $\hat{x} \in [0, \frac{D + q_X - q_Y}{2(1 - q_Y)}]$,

$$\underline{\theta} = \frac{q_Y - q_X}{1 - \hat{x}},$$
$$\overline{\theta} = \frac{1 - q_X}{1 - \hat{x}}.$$

2. For $\hat{x} \in (\frac{D + q_X - q_Y}{2(1 - q_Y)}, \frac{q_X + q_Y - D}{2q_Y}]$,

$$\underline{\theta} = \frac{q_Y - q_X}{1 - \hat{x}},$$
$$\overline{\theta} = \frac{D - (1 - 2\hat{x})(q_Y - q_X)}{2\hat{x}(1 - \hat{x})}.$$

3. For $\hat{x} \in (\frac{q_X + q_Y - D}{2q_Y}, \frac{q_X}{q_Y}]$,

$$\underline{\theta} = \frac{q_Y - q_X}{1 - \hat{x}},$$
$$\overline{\theta} = \frac{q_X}{\hat{x}}.$$

4. For $\hat{x} > \frac{q_X}{q_Y}$, $[\underline{\theta}, \overline{\theta}]$ is empty.

Note that when $\hat{x} \in [0, \frac{q_X}{q_Y}]$ and $\theta = \underline{\theta}$,

$$(1 - q_X - (1 - \hat{x})\theta) \log \frac{1}{1 - q_X - (1 - \hat{x})\theta} + \theta \log \frac{1}{\theta} + (q_X - \hat{x}\theta) \log \frac{1}{q_X - \hat{x}\theta}$$

$$= (1 - q_Y) \log \frac{1}{1 - q_Y} + \underline{\theta} \log \frac{1}{\underline{\theta}} + (q_X - \hat{x}\underline{\theta}) \log \frac{1}{q_X - \hat{x}\underline{\theta}}$$

$$\geq H_b(q_Y)$$

$$= H(Y).$$

So it suffices to consider the case $\hat{x} \in [0, \frac{q_X}{q_Y}]$ and $\theta = \overline{\theta}$, for which (P'') is reduced to the following form:

$$\min_{\hat{x} \in [0, \frac{q_X}{q_Y}]} \eta(\hat{x}),$$

where

$$\eta(\hat{x}) \triangleq \begin{cases} \frac{1 - q_X}{1 - \hat{x}} \log \frac{1 - \hat{x}}{1 - q_X} + \frac{q_X - \hat{x}}{1 - \hat{x}} \log \frac{1 - \hat{x}}{q_X - \hat{x}}, & \hat{x} \in [0, \frac{D + q_X - q_Y}{2(1 - q_Y)}], \\ \frac{2\hat{x} - q_X + (1 - 2\hat{x})q_Y - D}{2\hat{x}} \log \frac{2\hat{x}}{2\hat{x} - q_X + (1 - 2\hat{x})q_Y - D} \\ \quad + \frac{D - (1 - 2\hat{x})(q_Y - q_X)}{2\hat{x}(1 - \hat{x})} \log \frac{2\hat{x}(1 - \hat{x})}{D - (1 - 2\hat{x})(q_Y - q_X)} \\ \quad + \frac{q_X + (1 - 2\hat{x})q_Y - D}{2(1 - \hat{x})} \log \frac{2(1 - \hat{x})}{q_X + (1 - 2\hat{x})q_Y - D}, & \hat{x} \in (\frac{D + q_X - q_Y}{2(1 - q_Y)}, \frac{q_X + q_Y - D}{2q_Y}], \\ \frac{\hat{x} - q_X}{\hat{x}} \log \frac{\hat{x}}{\hat{x} - q_X} + \frac{q_X}{\hat{x}} \log \frac{\hat{x}}{q_X}, & \hat{x} \in (\frac{q_X + q_Y - D}{2q_Y}, \frac{q_X}{q_Y}]. \end{cases}$$

Note that $\eta(\hat{x})$ is a continuous function over $[0, \frac{q_X}{q_Y}]$. Moreover, $\eta(\hat{x})$ decreases monotonically from $H_b(q_X)$ to $H_b(\frac{D_{\max}^{(B)} - D}{2 - q_X - q_Y - D})$ as $\hat{x}$ varies from 0 to $\frac{D + q_X - q_Y}{2(1 - q_Y)}$. Since $\eta(\hat{x})$ is a concave function of $\frac{q_X}{\hat{x}}$ for $\hat{x} \in [\frac{q_X + q_Y - D}{2q_Y}, \frac{q_X}{q_Y}]$, it follows that

$$\min_{\hat{x} \in [\frac{q_X + q_Y - D}{2q_Y}, \frac{q_X}{q_Y}]} \eta(\hat{x}) = \min\left\{\eta(\frac{q_X + q_Y - D}{2q_Y}), \eta(\frac{q_X}{q_Y})\right\} = \min\left\{H_b(\frac{2q_X q_Y}{q_X + q_Y - D}), H_b(q_Y)\right\}.$$

So we have

$$\min_{\hat{x} \in [0, \frac{D + q_X - q_Y}{2(1 - q_Y)}] \cup [\frac{q_X + q_Y - D}{2q_Y}, \frac{q_X}{q_Y}]} \eta(\hat{x})$$

$$= \min\left\{H_b(\frac{D_{\max}^{(B)} - D}{2 - q_X - q_Y - D}), H_b(\frac{2q_X q_Y}{q_X + q_Y - D}), H_b(q_Y)\right\}$$

$$= \min\left\{H_b(\frac{D_{\max}^{(B)} - D}{2 - q_X - q_Y - D}), H_b(\frac{2q_X q_Y}{q_X + q_Y - D})\right\} \tag{34}$$

$$= \min\left\{\eta(\frac{D + q_X - q_Y}{2(1 - q_Y)}), \eta(\frac{q_X + q_Y - D}{2q_Y})\right\}$$

$$\geq \min_{\hat{x} \in [\frac{D + q_X - q_Y}{2(1 - q_Y)}, \frac{q_X + q_Y - D}{2q_Y}]} \eta(\hat{x}),$$

where (34) is due to the fact that $H_b(q_Y) \geq H_b(q_X) \geq H_b(\frac{D_{\max}^{(B)} - D}{2 - q_X - q_Y - D})$ (with the first inequality being a consequence of $q_X \leq q_Y$ and $q_X + q_Y \leq 1$). So the problem boils down to solving

$$\min_{\hat{x} \in [\frac{D + q_X - q_Y}{2(1 - q_Y)}, \frac{q_X + q_Y - D}{2q_Y}]} \eta(\hat{x}).$$

It can be verified that the minimum is attained at $\hat{x} = \frac{D + q_X - q_Y}{2(1 - q_Y)}$. This completes the proof of (22). A graphical illustration of the entropy-constrained optimal transport plan for the binary case can be found in Figure 6.

*Proof of (23).* Note that

$$
D_{\mathrm{mse}}(\mathcal{B}(q_X), R) = \frac{1}{2} D_{\mathrm{ncr}}(\mathcal{B}(q_X), \mathcal{B}(q_X), R)
$$

$$
= \begin{cases} -\frac{(1-q_X)^2}{1-H_b^{-1}(R)} + 1 - q_X, & q_X \le \frac{1}{2}, \\ -\frac{q_X^2}{1-H_b^{-1}(R)} + q_X, & q_X > \frac{1}{2}, \end{cases} \quad R \in [0, H_b(q_X)],
$$

$$
D_{\mathrm{mse}}(\mathcal{B}(q_Y), R) = \frac{1}{2} D_{\mathrm{ncr}}(\mathcal{B}(q_Y), \mathcal{B}(q_Y), R)
$$

$$
= \begin{cases} -\frac{(1-q_Y)^2}{1-H_b^{-1}(R)} + 1 - q_Y, & q_Y \le \frac{1}{2}, \\ -\frac{q_Y^2}{1-H_b^{-1}(R)} + q_Y, & q_Y > \frac{1}{2}, \end{cases} \quad R \in [0, H_b(q_Y)].
$$

Moreover, we have

$$
\begin{cases} p_{\hat{X}^*}\left(\frac{q_X - H_b^{-1}(R)}{1-H_b^{-1}(R)}\right) = 1 - H_b^{-1}(R), \\ p_{\hat{X}^*}(1) = H_b^{-1}(R), \end{cases} \quad R \in [0, H_b(q_X)], q_X \le \frac{1}{2},
$$

$$
\begin{cases} p_{\hat{X}^*}(0) = H_b^{-1}(R), \\ p_{\hat{X}^*}\left(\frac{q_X}{1-H_b^{-1}(R)}\right) = 1 - H_b^{-1}(R), \end{cases} \quad R \in [0, H_b(q_X)], q_X \ge \frac{1}{2},
$$

$$
\begin{cases} p_{\hat{Y}^*}\left(\frac{q_Y - H_b^{-1}(R)}{1-H_b^{-1}(R)}\right) = 1 - H_b^{-1}(R), \\ p_{\hat{Y}^*}(1) = H_b^{-1}(R), \end{cases} \quad R \in [0, H_b(q_Y)], q_Y \le \frac{1}{2},
$$

$$
\begin{cases} p_{\hat{Y}^*}(0) = H_b^{-1}(R), \\ p_{\hat{Y}^*}\left(\frac{q_Y}{1-H_b^{-1}(R)}\right) = 1 - H_b^{-1}(R), \end{cases} \quad R \in [0, H_b(q_Y)], q_Y \ge \frac{1}{2}.
$$

So

$$
W_2^2(p_{\hat{X}^*}, p_{\hat{Y}^*})
$$

$$
= \begin{cases} \frac{(q_X - q_Y)^2}{1-H_b^{-1}(R)}, & q_X, q_Y \le \frac{1}{2} \text{ or } q_X, q_Y \ge \frac{1}{2}, \\ \frac{(q_Y - q_X + H_b^{-1}(R))^2}{1-H_b^{-1}(R)} + H_b^{-1}(R) - \frac{2q_Y(1-q_X)H_b^{-1}(R)}{(1-H_b^{-1}(R))^2}, & q_X < \frac{1}{2}, q_Y > \frac{1}{2}, \\ \frac{(q_X - q_Y + H_b^{-1}(R))^2}{1-H_b^{-1}(R)} + H_b^{-1}(R) - \frac{2q_X(1-q_Y)H_b^{-1}(R)}{(1-H_b^{-1}(R))^2}, & q_X > \frac{1}{2}, q_Y < \frac{1}{2}. \end{cases}
$$

Based on the above expressions, one can easily verify (23) and (24). In particular, it is worth noting that when $q_X, q_Y \le \frac{1}{2}$ or $q_X, q_Y \ge \frac{1}{2}$,

$$
\overline{D}_{\mathrm{ncr}}(\mathcal{B}(q_X), \mathcal{B}(q_Y), R) = D_{\mathrm{ncr}}(\mathcal{B}(q_X), \mathcal{B}(q_Y), R), \quad R \in [0, \min\{H_b(q_X), H_b(q_Y)\}],
$$

i.e., there is no penalty for using optimal quantizer and dequantizer in the conventional rate-distortion sense.

Remark: It is easy to verify that

$$
D_{\mathrm{cr}}(\mathcal{B}(q_X), R) \triangleq \min_{q_Y \in [0,1]} D_{\mathrm{cr}}(\mathcal{B}(q_X), \mathcal{B}(q_Y), R)
$$

$$
= \begin{cases} q_X \left(1 - \frac{R}{H_b(q_X)}\right), & R \in [0, H_b(q_X)], q_X \le \frac{1}{2}, \\ (1 - q_X) \left(1 - \frac{R}{H_b(q_X)}\right), & R \in [0, H_b(q_X)], q_X > \frac{1}{2}, \end{cases}
$$

$$
D_{\mathrm{ncr}}(\mathcal{B}(q_X), R) \triangleq \min_{q_Y \in [0,1]} D_{\mathrm{ncr}}(\mathcal{B}(q_X), \mathcal{B}(q_Y), R)
$$

$$
= \begin{cases} q_X - H_b^{-1}(R), & R \in [0, H_b(q_X)], q_X \le \frac{1}{2}, \\ 1 - q_X - H_b^{-1}(R), & R \in [0, H_b(q_X)], q_X > \frac{1}{2}, \end{cases}
$$

which are respectively the conventional one-shot distortion-distortion function (or equivalently, one-shot distortion-rate-perception function with an inactive perception constraint) for $\mathcal{B}(q_X)$ with and without common randomness. In general, $D_{\mathrm{ncr}}(\mathcal{B}(q_X), R)$ is different from $D_{\mathrm{mse}}(\mathcal{B}(q_X), R)$ (the former is strictly above the latter). The reason is as follows: even though the output distribution constraint is removed in the definition of $D_{\mathrm{ncr}}(\mathcal{B}(q_X), R)$, the output alphabet remains to be $\{0, 1\}$; in contrast, for $D_{\mathrm{mse}}(\mathcal{B}(q_X), R)$, the output alphabet is relaxed to $\mathbb{R}$. □

## A.5 AUXILIARY RESULTS

**Lemma 1** (Finite Support Approximation). For any $R \geq 0$ and $\epsilon > 0$, there exists a joint distribution $p_{X,\hat{X},\hat{Y},Y}$ compatible with the given marginal distributions $p_X$ and $p_Y$ such that $X \leftrightarrow \hat{X} \leftrightarrow \hat{Y} \leftrightarrow Y$ form a Markov chain and $\hat{X}$ is a finite-support random variable with $H(\hat{X}) \leq R$ (or $\hat{Y}$ is a finite-support random variable with $H(\hat{Y}) \leq R$); moreover, $\mathbb{E}[X|\hat{X}] = \hat{X}$, $\mathbb{E}[Y|\hat{Y}] = \hat{Y}$, and

$$\mathbb{E}[\|X - \hat{X}\|^2] + \mathbb{E}[\|Y - \hat{Y}\|] + \mathbb{E}[\|\hat{X} - \hat{Y}\|^2] \leq D_{\mathrm{ncr}}(p_X, p_Y, R) + \epsilon.$$

*Proof.* In light of Theorem 1, we can find $p_{X,\hat{X},\hat{Y},Y}$ such that $X \leftrightarrow \hat{X} \leftrightarrow \hat{Y} \leftrightarrow Y$ form a Markov chain, $H(\hat{X}) \leq R$, $\mathbb{E}[X|\hat{X}] = \hat{X}$, $\mathbb{E}[Y|\hat{Y}]$, and

$$\mathbb{E}[\|X - \hat{X}\|^2] + \mathbb{E}[\|Y - \hat{Y}\|] + \mathbb{E}[\|\hat{X} - \hat{Y}\|^2] \leq D_{\mathrm{ncr}}(p_X, p_Y, R) + \frac{\epsilon}{2}. \tag{35}$$

The proof is complete if $\hat{X}$ is a finite-support random variable. So it suffices to consider the case where $\hat{X}$ takes value from some countably infinite set $\{\hat{x}_i\}_{i=1}^{\infty}$. Since $\mathbb{E}[\|X\|^2] < \infty$ and $\mathbb{E}[\|Y\|^2] < \infty$, it follows that there exists a positive integer $N$ such that

$$\mathbb{P}\{\hat{X} \in \{\hat{x}_i\}_{i=N}^{\infty}\}\mathbb{E}[\|X\|^2 + \|Y\|^2|\hat{X} \in \{\hat{x}_i\}_{i=N}^{\infty}] \leq \frac{\epsilon}{4}.$$

Let $\hat{X}' \triangleq \hat{X}$ if $\hat{X} \in \{\hat{x}_i\}_{i=1}^{N-1}$ and $\hat{X}' \triangleq \mathbb{E}[X|\hat{X} \in \{\hat{x}_i\}_{i=N}^{\infty}]$ if $\hat{X} \in \{\hat{x}_i\}_{i=N}^{\infty}$. Note that

$$
\begin{aligned}
&\mathbb{E}[\|X - \hat{X}'\|^2] + \mathbb{E}[\|\hat{X}' - \hat{Y}\|^2] \\
&= \mathbb{P}\{\hat{X} \in \{\hat{x}_i\}_{i=1}^{N-1}\}\mathbb{E}[\|X - \hat{X}'\|^2 + \|\hat{X}' - \hat{Y}\|^2|\hat{X} \in \{\hat{x}_i\}_{i=1}^{N-1}] \\
&\quad + \mathbb{P}\{\hat{X} \in \{\hat{x}_i\}_{i=N}^{\infty}\}\mathbb{E}[\|X - \hat{X}'\|^2 + \|\hat{X}' - \hat{Y}\|^2|\hat{X} \in \{\hat{x}_i\}_{i=N}^{\infty}] \\
&= \mathbb{P}\{\hat{X} \in \{\hat{x}_i\}_{i=1}^{N-1}\}\mathbb{E}[\|X - \hat{X}\|^2 + \|\hat{X} - \hat{Y}\|^2|\hat{X} \in \{\hat{x}_i\}_{i=1}^{N-1}] \\
&\quad + \mathbb{P}\{\hat{X} \in \{\hat{x}_i\}_{i=N}^{\infty}\}\mathbb{E}[\|X - \hat{X}'\|^2 + \|\hat{X}' - \hat{Y}\|^2|\hat{X} \in \{\hat{x}_i\}_{i=N}^{\infty}] \\
&\leq \mathbb{P}\{\hat{X} \in \{\hat{x}_i\}_{i=1}^{N-1}\}\mathbb{E}[\|X - \hat{X}\|^2 + \|\hat{X} - \hat{Y}\|^2|\hat{X} \in \{\hat{x}_i\}_{i=1}^{N-1}] \\
&\quad + \mathbb{P}\{\hat{X} \in \{\hat{x}_i\}_{i=N}^{\infty}\}\mathbb{E}[\|X - \hat{X}'\|^2 + 2\|\hat{X}'\|^2 + 2\|\hat{Y}\|^2|\hat{X} \in \{\hat{x}_i\}_{i=N}^{\infty}] \\
&\leq \mathbb{P}\{\hat{X} \in \{\hat{x}_i\}_{i=1}^{N-1}\}\mathbb{E}[\|X - \hat{X}\|^2 + \|\hat{X} - \hat{Y}\|^2|\hat{X} \in \{\hat{x}_i\}_{i=1}^{N-1}] \\
&\quad + \mathbb{P}\{\hat{X} \in \{\hat{x}_i\}_{i=N}^{\infty}\}\mathbb{E}[\|X\|^2 + \|\hat{X}'\|^2 + 2\|\hat{Y}\|^2|\hat{X} \in \{\hat{x}_i\}_{i=N}^{\infty}] \\
&\leq \mathbb{E}[\|X - \hat{X}\|^2 + \|\hat{X} - \hat{Y}\|^2] + 2\mathbb{P}\{\hat{X} \in \{\hat{x}_i\}_{i=N}^{\infty}\}\mathbb{E}[\|X\|^2 + \|Y\|^2|\hat{X} \in \{\hat{x}_i\}_{i=N}^{\infty}] \\
&\leq \mathbb{E}[\|X - \hat{X}\|^2] + \mathbb{E}[\|\hat{X} - \hat{Y}\|^2] + \frac{\epsilon}{2}. \tag{36}
\end{aligned}
$$

Now define a new joint distribution $p_{X,\hat{X}'',\hat{Y}'',Y}$ such that $p_{X,\hat{X}''} = p_{X,\hat{X}'}$, $p_{\hat{X}'',\hat{Y}''} = p_{\hat{X}',\hat{Y}}$, $p_{Y,\hat{Y}''} = p_{Y,\hat{Y}}$, and $X \leftrightarrow \hat{X}'' \leftrightarrow \hat{Y}'' \leftrightarrow Y$ form a Markov chain. It is clear that $\hat{X}''$ is a finite-support random variable with $H(\hat{X}'') = H(\hat{X}') \leq H(\hat{X}) \leq R$, $\mathbb{E}[X|\hat{X}''] = \hat{X}''$, $\mathbb{E}[Y|\hat{Y}''] = \hat{Y}''$, and

$$
\begin{aligned}
&\mathbb{E}[\|X - \hat{X}''\|^2] + \mathbb{E}[\|Y - \hat{Y}''\|] + \mathbb{E}[\|\hat{X}'' - \hat{Y}''\|^2] \\
&\leq \mathbb{E}[\|X - \hat{X}'\|^2] + \mathbb{E}[\|Y - \hat{Y}\|] + \mathbb{E}[\|\hat{X}' - \hat{Y}\|^2] \\
&\leq \mathbb{E}[\|X - \hat{X}\|^2] + \mathbb{E}[\|Y - \hat{Y}\|] + \mathbb{E}[\|\hat{X} - \hat{Y}\|^2] + \frac{\epsilon}{2} \tag{37} \\
&\leq D_{\mathrm{ncr}}(p_X, p_Y, R) + \epsilon, \tag{38}
\end{aligned}
$$

where (37) and (38) are due to (35) and (36), respectively. This completes the proof of Lemma 1. $\quad\square$

**Lemma 2** (Deterministic on Finite Support). Let $X \leftrightarrow \hat{X} \leftrightarrow \hat{Y} \leftrightarrow Y$ be a Markov chain with $\mathbb{E}[X|\hat{X}] = \hat{X}$, $\mathbb{E}[Y|\hat{Y}] = \hat{Y}$, and assume that $\hat{X}$ (or $\hat{Y}$) is a finite-support random variable. There exist deterministically related random variables $\hat{X}'$ and $\hat{Y}'$, with the support size no greater than that of

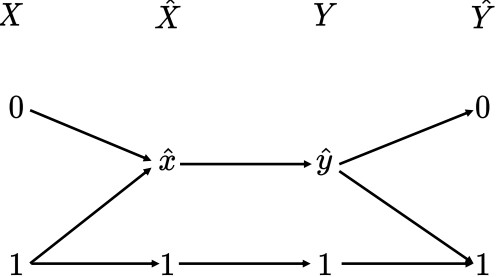

Figure 6: Illustration of the entropy-constrained optimal transport plan for the binary case (assuming $q_X + q_Y \leq 1$), where $p_{\hat{X}}(\hat{x}) = p_{\hat{Y}}(\hat{y}) = 1 - H_b^{-1}(R)$ with $\hat{x} = \frac{q_X - H_b^{-1}(R)}{1 - H_b^{-1}(R)}$ and $\hat{y} = \frac{q_Y - H_b^{-1}(R)}{1 - H_b^{-1}(R)}$. It is interesting to note that the quantizer $p_{\hat{X}|X}$ does not depend on $p_Y$ while the dequantizer $p_{Y|\hat{Y}}$ does not depend on $p_X$. So they are decoupled in a certain sense. Moreover, $p_{\hat{X}|X}$ and $p_{Y|\hat{Y}}$ coincide respectively with optimal quantizer $p_{\hat{X}^*|X}$ and dequantizer $p_{Y|\hat{Y}^*}$ in the conventional rate-distortion sense when $q_X, q_Y \leq 1/2$.

$\hat{X}$ and $H(\hat{X}') = H(\hat{Y}') \leq H(\hat{X})$ (or $H(\hat{X}') = H(\hat{Y}') \leq H(\hat{Y})$), such that $X \leftrightarrow \hat{X}' \leftrightarrow \hat{Y}' \leftrightarrow Y$ form a Markov chain, $\mathbb{E}[X|\hat{X}'] = \hat{X}'$, $\mathbb{E}[Y|\hat{Y}'] = \hat{Y}'$, and

$$\mathbb{E}[\|X - \hat{X}'\|^2] + \mathbb{E}[\|Y - \hat{Y}'\|] + \mathbb{E}[\|\hat{X}' - \hat{Y}'\|^2]$$
$$= \mathbb{E}[\|X - \hat{X}\|^2] + \mathbb{E}[\|Y - \hat{Y}\|] + \mathbb{E}[\|\hat{X} - \hat{Y}\|^2].$$

*Proof.* Let $\tilde{Y} \triangleq \mathbb{E}[Y|\hat{X}]$. Since $\tilde{Y} \leftrightarrow \hat{Y} \leftrightarrow Y$ form a Markov chain and $\mathbb{E}[Y|\hat{Y}] = \hat{Y}$, we have $\tilde{Y} = \mathbb{E}[\hat{Y}|\hat{X}]$. Construct a new joint distribution $p_{X,\hat{X}',\hat{Y}',Y}$ such that $p_{X,\hat{X}'} = p_{X,\hat{X}}$, $p_{\hat{X}',\hat{Y}'} = p_{\hat{X},\tilde{Y}}$, $p_{Y,\hat{Y}'} = p_{Y,\tilde{Y}}$, and $X \leftrightarrow \hat{X}' \leftrightarrow \hat{Y}' \leftrightarrow Y$ form a Markov chain. It is clear that $\mathbb{E}[X|\hat{X}'] = \hat{X}'$, $\mathbb{E}[Y|\hat{Y}'] = \hat{Y}'$, and

$$\mathbb{E}[\|X - \hat{X}'\|^2] + \mathbb{E}[\|Y - \hat{Y}'\|^2] + \mathbb{E}[\|\hat{X}' - \hat{Y}'\|^2]$$
$$= \mathbb{E}[\|X - \hat{X}\|^2] + \mathbb{E}[\|Y - \tilde{Y}\|^2] + \mathbb{E}[\|\hat{X} - \tilde{Y}\|^2]$$
$$= \mathbb{E}[\|X - \hat{X}\|^2] + \mathbb{E}[\|Y - \hat{Y}\|^2] + \mathbb{E}[\|\hat{Y} - \tilde{Y}\|^2] + \mathbb{E}[\|\hat{X} - \tilde{Y}\|^2]$$
$$= \mathbb{E}[\|X - \hat{X}\|^2] + \mathbb{E}[\|Y - \hat{Y}\|^2] + \mathbb{E}[\|\hat{X} - \hat{Y}\|^2],$$

where the last equality is due to $\tilde{Y} = \mathbb{E}[\hat{Y}|\hat{X}]$. If the function that maps $\hat{X}'$ to $\hat{Y}'$ (or equivalently, maps $\hat{X}$ to $\tilde{Y}$) is invertible, then $\hat{X}$, $\hat{X}'$, $\hat{Y}'$ have the same support size and $H(\hat{X}) = H(X') = H(Y')$, which completes the proof. Otherwise, the support size of $\hat{Y}'$ must be strictly smaller than that of $\hat{X}'$ (which is the same as that of $\hat{X}$) and $H(\hat{Y}') < H(\hat{X}') = H(\hat{X})$. We can alternately reduce the support sizes of $\hat{X}'$ and $\hat{Y}'$ using this argument until they are deterministically related (and consequently have the same support size and the same entropy). This can be accomplished in a finite number of steps because the reduction in support size cannot continue forever. □

□

### A.6 UNIFORM DISTRIBUTION

Let $X \sim \text{Unif}[0, a]$ and $Y \sim \text{Unif}[0, b]$ be uniformly distributed random variables, where $a, b > 0$. Note that the density functions are given as: $p_X(x) = \frac{1}{a}, 0 \leq x \leq a$ and $p_Y(y) = \frac{1}{b}, 0 \leq y \leq b$ and $p_X(x)$ and $p_Y(y)$ are zero outside these intervals. The cumulative density functions of $X$ and $Y$ are

given as follows:

$$C_X(x) = \begin{cases} 0, & x \le 0, \\ \frac{x}{a}, & 0 \le x \le a, \\ 1, & x \ge a, \end{cases} \qquad C_Y(y) = \begin{cases} 0, & y \le 0, \\ \frac{y}{b}, & 0 \le y \le b, \\ 1, & y \ge b. \end{cases}$$

Following Peyré & Cuturi (2019, Remark 2.30) we have that the optimal transport (without rate constraint) is given by:

$$W_2^2(p_X, p_Y) = \int_0^1 (C_X^{-1}(r) - C_Y^{-1}(r))^2 dr = \frac{(b-a)^2}{3}, \tag{39}$$

where $C_X^{-1}(\cdot)$ and $C_Y^{-1}(\cdot)$ are the pseudo-inverse of the CDF functions for $X$ and $Y$ as defined in Peyré & Cuturi (2019, Remark 2.30).

We will next develop an upper bound on $D_{\mathrm{ncr}}(p_X, p_Y, R)$ using Theorem 1 when the rate is of the form $R = \log_2(N)$ for any $N \in \{1, 2, \ldots\}$, by considering the following choice for $\hat{X}$ and $\hat{Y}$:

$$\hat{X} \in \hat{\mathcal{X}} = \left\{ \frac{a}{2N}, \frac{3a}{2N}, \frac{5a}{2N}, \ldots, \frac{(2N-1)a}{2N} \right\}, \tag{40}$$

$$\hat{Y} \in \hat{\mathcal{Y}} = \left\{ \frac{b}{2N}, \frac{3b}{2N}, \frac{5b}{2N}, \ldots, \frac{(2N-1)b}{2N} \right\}. \tag{41}$$

To compute the upper bound we select $p_{\hat{X}|X}$ to correspond to scalar quantization of $X$ i.e., given $X$ we select $\hat{X}$ as a point in $\hat{\mathcal{X}}$ closest to $X$. The distribution $p_{\hat{Y}|Y}$ is defined in an analogous manner[3]. Our upper bound can be computed as:

$$D_{\mathrm{ncr}}^+(p_X, p_Y, R) = \mathbb{E}[(X - \hat{X})^2] + \mathbb{E}[(Y - \hat{Y})^2] + W_2^2(p_{\hat{X}}, p_{\hat{Y}}). \tag{42}$$

Note that with $\Delta = \frac{1}{N}$ we have that $\mathbb{E}[(X - \hat{X})^2] = \frac{a^2 \Delta^2}{12}$ and $\mathbb{E}[(Y - \hat{Y})^2] = \frac{b^2 \Delta^2}{12}$. Thus we only need to compute the third term. Following Peyré & Cuturi (2019, Remark 2.28) we have that:

$$W_2^2(p_{\hat{X}}, p_{\hat{Y}}) = \frac{(b-a)^2 \Delta^2}{4N} \sum_{i=1}^N (2i-1)^2 = \frac{(b-a)^2}{3} \left( 1 - \frac{\Delta^2}{4} \right). \tag{43}$$

Thus using $\Delta^2 = 2^{-2R}$, we have that

$$D_{\mathrm{ncr}}^+(p_X, p_Y, R) = \frac{(b-a)^2}{3} + \frac{a \cdot b}{6} 2^{-2R}. \tag{44}$$

Note that the upper bound approaches the lower bound in (39) as $R \to \infty$, with exponential rate of convergence.

For the case of general $R$, we let $N = \lceil 2^R \rceil$ and following Gyorgy & Linder (2000, Theorem 1), we select

$$\hat{X} \in \hat{\mathcal{X}} = \left\{ \underbrace{\frac{ac}{2}}_{\hat{x}_1}, \underbrace{a\left(c + \frac{c'}{2}\right)}_{\hat{x}_2}, \underbrace{a\left(c + 3\frac{c'}{2}\right)}_{\hat{x}_3}, \ldots, \underbrace{a\left(c + (2N-3)\frac{c'}{2}\right)}_{\hat{x}_N} \right\},$$

$$\hat{Y} \in \hat{\mathcal{Y}} = \left\{ \underbrace{\frac{bc}{2}}_{\hat{y}_1}, \underbrace{b\left(c + \frac{c'}{2}\right)}_{\hat{y}_2}, \underbrace{b\left(c + 3\frac{c'}{2}\right)}_{\hat{y}_3}, \ldots, \underbrace{b\left(c + (2N-3)\frac{c'}{2}\right)}_{\hat{y}_N} \right\}, \tag{45}$$

---

[3]Please note that we do not claim that the proposed choice is optimal with respect to $D_{\mathrm{mse}}$ in (4) although it is known to be optimal solution for a related problem - the entropy constrained scalar quantization (Gyorgy & Linder (2000)). As a result we cannot claim to compute the upper bound $\bar{D}_{\mathrm{ncr}}$ stated Theorem 1 but provide another upper bound.

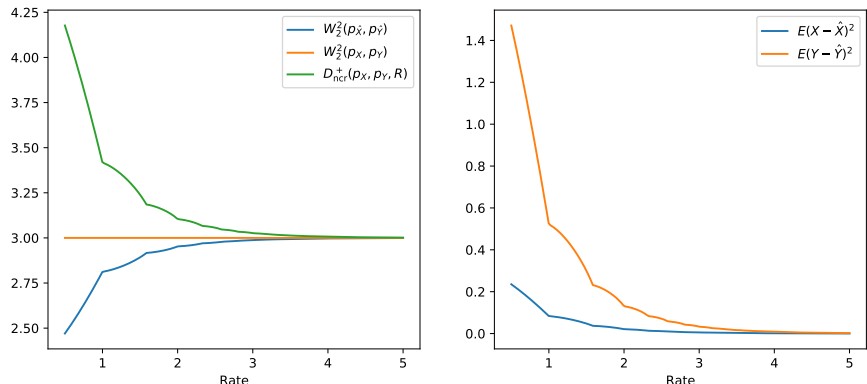

Figure 7: Example of Uniform sources with $a = 2$ and $b = 5$. The left plot shows the lower bound $W_2^2(p_X, p_Y)$ in (39) and the upper bound $D_{\text{ncr}}^+(p_X, p_Y, R)$ which is the sum of of the right hand side in (46), (47) and (50). For comparison we also show the value of $W_2^2(p_{\hat{X}}, p_{\hat{Y}})$. The right plot shows the distortions associated with the quantization and dequantization steps.

where $c$ is the unique solution in the interval $(0, 1/N]$ to the equation:

$$-c \log c - (1 - c) \log \frac{(1 - c)}{N - 1} = R,$$

and $c' = \frac{(1-c)}{N-1}$ holds. Note that the length of the first interval is $c$ and the length of all other intervals is $c'$. In the special case where $R = \log_2 N$ we will have that $c = c' = \frac{1}{N}$ and our construction for $\hat{X}$ and $\hat{Y}$ is consistent with the previous case.

As before we use $p_{\hat{X}|X}$ and $p_{\hat{Y}|Y}$ to be the distributions associated with scalar quantization. Thus we have that:

$$\mathbb{E}[(X - \hat{X})^2] = c \frac{a^2 c^2}{12} + (1 - c) \frac{a^2 c'^2}{12}, \tag{46}$$

$$\mathbb{E}[(Y - \hat{Y})^2] = c \frac{b^2 c^2}{12} + (1 - c) \frac{b^2 c'^2}{12}. \tag{47}$$

Furthermore using the result stated in Peyré & Cuturi (2019, Remark 2.30) we have that

$$W_2^2(p_{\hat{X}}, p_{\hat{Y}}) = c(\hat{x}_1 - \hat{y}_1)^2 + c' \sum_{j=2}^{N} (\hat{x}_j - \hat{y}_j)^2 \tag{48}$$

$$= (b - a)^2 \frac{c^3}{4} + (b - a)^2 c' \sum_{j=1}^{N-1} \left( c + \frac{2j - 1}{2} c' \right)^2 \tag{49}$$

$$= (b - a)^2 \left( \frac{c^3}{4} + c^2 c'(N - 1) + cc'^2(N - 1)^2 + \frac{1}{12} c'^3(2N - 1)(2N - 3)(N - 1) \right). \tag{50}$$

Finally, the upper bound $D_{\text{ncr}}^+(p_X, p_Y, R)$ can be obtained by summing the right hand side of (46), (47) and (50). We provide a numerical evaluation of this upper bound in Fig. 7.

## A.7 ASYMPTOTIC OPTIMAL TRANSPORT

Let $X_1, X_2, \cdots$ and $Y_1, Y_2, \cdots$ be i.i.d. processes with marginal distributions $p_X$ and $p_Y$, respectively.

**Definition 4** (Asymptotic Optimal Transport with Entropy Bottleneck — no common randomness). The asymptotic optimal transport from $p_X$ to $p_Y$ with an entropy bottleneck of $R$ and without common randomness is defined as

$$D_{\mathrm{ncr}}^{(\infty)}(p_X, p_Y, R) \triangleq \inf_{n \geq 1} D_{\mathrm{ncr}}^{(n)}(p_X, p_Y, R),$$

where

$$D_{\mathrm{ncr}}^{(n)}(p_X, p_Y, R) \triangleq \inf_{p_{X^n, Z, Y^n} \in M_{\mathrm{ncr}}(\otimes_{i=1}^n p_X, \otimes_{i=1}^n p_Y)} \frac{1}{n} \sum_{i=1}^n \mathbb{E}[d(X_i, Y_i)]$$

$$\text{s.t.} \quad \frac{1}{n} H(Z) \leq R.$$

Remark: It is clear that $D_{\mathrm{ncr}}^{(1)}(p_X, p_Y, R) = D_{\mathrm{ncr}}(p_X, p_Y, R)$. Moreover, one can readily show that $\{n D_{\mathrm{ncr}}^{(n)}(p_X, p_Y, R)\}_{n=1}^{\infty}$ is a subadditive sequence and consequently $D_{\mathrm{ncr}}^{(\infty)}(p_X, p_Y, R) = \lim_{n \to \infty} D_{\mathrm{ncr}}^{(n)}(p_X, p_Y, R)$.

**Theorem 5.** *We have*

$$D_{\mathrm{ncr}}^{(\infty)}(p_X, p_Y, R) = \inf_{p_{X,Z,Y} \in M_{\mathrm{ncr}}(p_X, p_Y)} \mathbb{E}[d(X, Y)]$$

$$\text{s.t.} \quad \max\{I(X; Z), I(Y; Z)\} \leq R.$$

*Proof.* This result can be specialized from Theorem 1 in Saldi et al. (2015b). $\quad\square$

The following result is the counterpart of Theorem 1 in the asymptotic setting.

**Theorem 6.** *Let*

$$D_{\mathrm{mse}}^{(\infty)}(p_X, p_Y, R) \triangleq \inf_{p_{\hat{X}|X}, p_{\hat{Y}|Y}} \mathbb{E}[\|X - \hat{X}\|^2] + \mathbb{E}[\|Y - \hat{Y}\|^2] + W_2^2(p_{\hat{X}}, p_{\hat{Y}}) \tag{51}$$

$$\text{s.t.} \quad \mathbb{E}[X|\hat{X}] = \hat{X}, \quad \mathbb{E}[Y|\hat{Y}] = \hat{Y}, \quad I(X; \hat{X}) \leq R, \quad I(Y; \hat{Y}) \leq R,$$

*and*

$$D_{\mathrm{mse}}(p_X, R) \triangleq \inf_{p_{\hat{X}|X}} \mathbb{E}[\|X - \hat{X}\|^2] \tag{52}$$

$$\text{s.t.} \quad I(X; \hat{X}) \leq R.$$

*Moreover, let*

$$\overline{D}_{\mathrm{ncr}}^{(\infty)}(p_X, p_Y, R) \triangleq D_{\mathrm{mse}}^{(\infty)}(p_X, R) + D_{\mathrm{mse}}^{(\infty)}(p_Y, R) + W_2^2(p_{\hat{X}^*}, p_{\hat{Y}^*}), \tag{53}$$

$$\underline{D}_{\mathrm{ncr}}^{(\infty)}(p_X, p_Y, R) \triangleq D_{\mathrm{mse}}^{(\infty)}(p_X, R) + D_{\mathrm{mse}}^{(\infty)}(p_Y, R), \tag{54}$$

*where $p_{\hat{X}^*}$ and $p_{\hat{Y}^*}$ are the marginal distributions induced by the minimizers $p_{\hat{X}^*|X}$ and $p_{\hat{Y}^*|Y}$ that attain $D_{\mathrm{mse}}^{(\infty)}(p_X, R)$ and $D_{\mathrm{mse}}^{(\infty)}(p_Y, R)$, respectively (assuming the existence and uniqueness of such minimizers). Then under the squared Eucledian distortion measure,*

$$D_{\mathrm{ncr}}^{(\infty)}(p_X, p_Y, R) = D_{\mathrm{mse}}^{(\infty)}(p_X, p_Y, R). \tag{55}$$

*In addition, we have*

$$\overline{D}_{\mathrm{ncr}}^{(\infty)}(p_X, p_Y, R) \geq D_{\mathrm{ncr}}^{(\infty)}(p_X, p_Y, R) \geq \underline{D}_{\mathrm{ncr}}^{(\infty)}(p_X, p_Y, R), \tag{56}$$

*and both inequalities are tight when $p_X = p_Y$.*

*Proof.* For any $p_{X,Z,Y} \in M_{\mathrm{ncr}}(p_X, p_Y)$ with $\max\{I(X; Z), I(Y; Z)\} \leq R$,

$$\mathbb{E}[\|X - Y\|^2] = \mathbb{E}[\|X - \mathbb{E}[X|Z]\|^2] + \mathbb{E}[\|Y - \mathbb{E}[Y|Z]\|^2] + \mathbb{E}[\|\mathbb{E}[X|Z] - \mathbb{E}[Y|Z]\|^2]$$

$$\geq D_{\text{mse}}^{(\infty)}(p_X, p_Y, R),$$

where the last inequality follows from the definition of $D_{\text{mse}}^{(\infty)}(p_X, p_Y, R)$ and the fact that

$$\max\{I(X; \mathbb{E}[X|Z]), I(Y; \mathbb{E}[Y|Z])\} \leq \max\{I(X; Z), I(Y; Z)\} \leq R.$$

In view of Theorem 5, we must have $D_{\text{ncr}}^{(\infty)}(p_X, p_Y, R) \geq D_{\text{mse}}^{(\infty)}(p_X, p_Y, R)$. On the other hand, for any $p_{\hat{X}|X}$, $p_{\hat{Y}|Y}$ with $\mathbb{E}[X|\hat{X}] = \hat{X}$, $\mathbb{E}[Y|\hat{Y}] = \hat{Y}$, $I(X; \hat{X}) \leq R$, and $I(Y; \hat{Y}) \leq R$, we can construct a joint distribution $p_{X, \hat{X}, \hat{Y}, Y}$ such that $X \leftrightarrow \hat{X} \leftrightarrow \hat{Y} \leftrightarrow Y$ form a Markov chain, $p_{X, \hat{X}} = p_X p_{\hat{X}|X}$, $p_{Y, \hat{Y}} = p_Y p_{\hat{Y}|Y}$, and $p_{\hat{X}, \hat{Y}}$ satisfying $\mathbb{E}[\|\hat{X} - \hat{Y}\|^2] = W_2^2(p_{\hat{X}}, p_{\hat{Y}})$. Note that

$$\mathbb{E}[\|X - Y\|^2] = \mathbb{E}[\|X - \hat{X}\|^2] + \mathbb{E}[\|Y - \hat{Y}\|^2] + \mathbb{E}[\|\hat{X} - \hat{Y}\|^2]$$
$$= \mathbb{E}[\|X - \hat{X}\|^2] + \mathbb{E}[\|Y - \hat{Y}\|^2] + W_2^2(p_{\hat{X}}, p_{\hat{Y}}). \tag{57}$$

Let $Z \triangleq \hat{X}$. It can be verified that $p_{X, Z, Y} \in M_{\text{ncr}}(p_X, p_Y)$ and $\max\{I(X; Z), I(Y; Z)\} = \max\{I(X; \hat{X}), I(Y; \hat{X})\} \leq \max\{I(X; \hat{X}), I(Y; \hat{Y})\} \leq R$, which, together with (57), implies $D_{\text{ncr}}^{(\infty)}(p_X, p_Y, R) \leq D_{\text{mse}}^{(\infty)}(p_X, p_Y, R)$. This completes the proof of (55).

Dropping the term $W_2^2(p_{\hat{X}}, p_{\hat{Y}})$ in (51) yields

$$D_{\text{ncr}}^{(\infty)}(p_X, p_Y, R) \geq \tilde{D}_{\text{mse}}^{(\infty)}(p_X, R) + \tilde{D}_{\text{mse}}^{(\infty)}(p_Y, R),$$

where

$$\tilde{D}_{\text{mse}}^{(\infty)}(p_X, R) \triangleq \inf_{p_{\hat{X}|X}} \mathbb{E}[\|X - \hat{X}\|^2]$$
$$\text{s.t.} \quad \mathbb{E}[X|\hat{X}] = \hat{X}, \quad I(X; \hat{X}) \leq R.$$

and $\tilde{D}_{\text{mse}}(p_Y, R)$ is definely analogously. On the other hand, choosing $p_{\hat{X}|X} = p_{\hat{X}'|X}$ and $p_{\hat{Y}|Y} = p_{\hat{Y}'|Y}$ in (51) gives

$$D_{\text{ncr}}^{(\infty)}(p_X, p_Y, R) \leq \tilde{D}_{\text{mse}}^{(\infty)}(p_X, R) + \tilde{D}_{\text{mse}}^{(\infty)}(p_Y, R) + W_2^2(p_{\hat{X}'}, p_{\hat{Y}'}),$$

where $p_{\hat{X}'|X}$ and $p_{\hat{Y}'|Y}$ are the minimizers that attain $\tilde{D}_{\text{mse}}^{(\infty)}(p_X, R)$ and $\tilde{D}_{\text{mse}}^{(\infty)}(p_Y, R)$ respectively while $p_{\hat{X}'}$ and $p_{\hat{Y}'}$ are their induced marginal distributions. It is clear that $p_{\hat{X}'|X}$ and $p_{\hat{Y}'|Y}$ coincide with $p_{\hat{X}^*|X}$ and $p_{\hat{Y}^*|Y}$ respectively as the constraints $\mathbb{E}[X|\hat{X}] = \hat{X}$ and $\mathbb{E}[Y|\hat{Y}] = \hat{Y}$ are automatically satisfied by $p_{\hat{X}^*|X}$ and $p_{\hat{Y}^*|Y}$. This proves (56). For the special case $p_X = p_Y$, we have $p_{\hat{X}^*|X} = p_{\hat{Y}^*|Y}$ and consequently the upper bound and the lower bound in (56) coincide. $\square$

**Definition 5** (Asymptotic Optimal Transport with Entropy Bottleneck — with Common Randomness). The asymptotic optimal transport from $p_X$ to $p_Y$ with entropy bottleneck $R$ and common randomness is defined as

$$D_{\text{cr}}^{(\infty)}(p_X, p_Y, R) \triangleq \inf_{n \geq 1} D_{\text{cr}}^{(n)}(p_X, p_Y, R),$$

where

$$D_{\text{cr}}^{(n)}(p_X, p_Y, R) \triangleq \inf_{p_{U, X^n, Z, Y^n} \in M_{\text{cr}}(\otimes_{i=1}^n p_X, \otimes_{i=1}^n p_Y)} \frac{1}{n} \sum_{i=1}^{n} \mathbb{E}[d(X_i, Y_i)]$$
$$\text{s.t.} \quad \frac{1}{n} H(Z|U) \leq R.$$

Remark: It is clear that $D_{\text{cr}}^{(1)}(p_X, p_Y, R) = D_{\text{cr}}(p_X, p_Y, R)$. Moreover, one can readily show that $\{n D_{\text{cr}}^{(n)}(p_X, p_Y, R)\}_{n=1}^{\infty}$ is a subadditive sequence and consequently $D_{\text{cr}}^{(\infty)}(p_X, p_Y, R) = \lim_{n \to \infty} D_{\text{ncr}}^{(n)}(p_X, p_Y, R)$.

**Theorem 7.** *We have*

$$D_{\text{cr}}^{(\infty)}(p_X, p_Y, R) = \inf_{p_{X,Y} \in \Gamma(p_X, p_Y)} \mathbb{E}[d(X, Y)]$$
$$s.t. \quad I(X; Y) \leq R.$$

*Proof.* This result is known (see Theorem 7 in Saldi et al. (2015a)). It is possible to give a simpler proof of the achievability part by leveraging Theorem 3 and the strong data processing inequality Li & El Gamal (2018). The converse is based on standard information-theoretic arguments. $\square$

## A.8 GAUSSIAN CASE

Let $X \sim \mathcal{N}(\mu_X, \sigma_X^2)$ and $Y \sim \mathcal{N}(\mu_Y, \sigma_Y^2)$ be two Gaussian random variables, and let $d(\cdot, \cdot)$ be the squared distortion measure (i.e., $d(x, y) = (x - y)^2$). Let $D_{\min}^{(G)} \triangleq (\mu_X - \mu_Y)^2 + (\sigma_X - \sigma_Y)^2$ and $D_{\max}^{(G)} \triangleq (\mu_X - \mu_Y)^2 + \sigma_X^2 + \sigma_Y^2$. Note that $D_{\min}^{(G)}$ is the squared Wasserstein-2 distance between $\mathcal{N}(\mu_X, \sigma_X^2)$ and $\mathcal{N}(\mu_Y, \sigma_Y)$, which is the minimum $\mathbb{E}[(X - Y)^2]$ achievable by coupling $X$ and $Y$. On the other hand, we have $\mathbb{E}[(X - Y)^2] = D_{\max}^{(G)}$ for $X, Y$ independent. It is clear that $D_{\min}^{(G)}$ and $D_{\max}^{(G)}$ are the infimum and supremum of $D_{\text{ncr}}^{(\infty)}(\mathcal{N}(\mu_X, \sigma_X^2), \mathcal{N}(\mu_Y, \sigma_Y^2), R)$ (as well as $D_{\text{cr}}^{(\infty)}(\mathcal{N}(\mu_X, \sigma_X^2), \mathcal{N}(\mu_Y, \sigma_Y^2), R)$), respectively.

**Theorem 8.** *Assume squared distortion measure. Under no common randomness, we have*

$$D_{\text{ncr}}^{(\infty)}(\mathcal{N}(\mu_X, \sigma_X^2), \mathcal{N}(\mu_Y, \sigma_Y^2), R) = D_{\min}^{(G)} + 2\sigma_X \sigma_Y 2^{-2R}, \quad R \in [0, \infty). \tag{58}$$

*Moreover,*

$$\overline{D}_{\text{ncr}}^{(\infty)}(\mathcal{N}(\mu_X, \sigma_X^2), \mathcal{N}(\mu_Y, \sigma_Y^2), R) = D_{\text{ncr}}^{(\infty)}(\mathcal{N}(\mu_X, \sigma_X^2), \mathcal{N}(\mu_Y, \sigma_Y^2), R), \quad R \in [0, \infty), \tag{59}$$
$$\underline{D}_{\text{ncr}}^{(\infty)}(\mathcal{N}(\mu_X, \sigma_X^2), \mathcal{N}(\mu_Y, \sigma_Y^2), R) = (\sigma_X^2 + \sigma_Y^2) 2^{-2R}, \quad R \in [0, \infty). \tag{60}$$

*With common randomness,*

$$D_{\text{cr}}^{(\infty)}(\mathcal{N}(\mu_X, \sigma_X^2), \mathcal{N}(\mu_Y, \sigma_Y^2), R) = D_{\max}^{(G)} - 2\sigma_X \sigma_Y \sqrt{1 - 2^{-2R}}, \quad R \in [0, \infty). \tag{61}$$

*Proof.* Consider $p_{\hat{X}|X}$ and $p_{\hat{Y}|Y}$ such that $\mathbb{E}[X|\hat{X}]$, $\mathbb{E}[Y|\hat{Y}]$, $I(X; \hat{X}) \leq R$, and $I(Y; \hat{Y}) \leq R$. Denote the mean and the variance of $\hat{X}$ by $\mu_{\hat{X}}$ and $\sigma_{\hat{X}}^2$, respectively. Similarly, denote the mean and the variance of $\hat{Y}$ by $\mu_{\hat{Y}}$ and $\sigma_{\hat{Y}}^2$, respectively. Clearly, $\mu_{\hat{X}} = \mu_X$, $\mu_{\hat{Y}} = \mu_Y$, $\sigma_{\hat{X}}^2 = \sigma_X^2 - \mathbb{E}[(X - \hat{X})^2]$, and $\sigma_{\hat{Y}}^2 = \sigma_Y^2 - \mathbb{E}[(Y - \hat{Y})^2]$. Moreover,

$$W_2^2(p_{\hat{X}}, p_{\hat{Y}}) \geq W_2^2(\mathcal{N}(\mu_{\hat{X}}, \sigma_{\hat{X}}^2), \mathcal{N}(\mu_{\hat{Y}}, \sigma_{\hat{Y}}^2))$$
$$= (\mu_{\hat{X}} - \mu_{\hat{Y}})^2 + (\sigma_{\hat{X}} - \sigma_{\hat{Y}})^2$$
$$= (\mu_X - \mu_Y)^2 + (\sigma_{\hat{X}} - \sigma_{\hat{Y}})^2.$$

So we have

$$\mathbb{E}[(X - \hat{X})^2] + \mathbb{E}[\|Y - \hat{Y}\|^2] + W_2^2(p_{\hat{X}}, p_{\hat{Y}})$$
$$\geq \sigma_X^2 - \sigma_{\hat{X}}^2 + \sigma_Y^2 - \sigma_{\hat{Y}}^2 + (\mu_X - \mu_Y)^2 + (\sigma_{\hat{X}} - \sigma_{\hat{Y}})^2$$
$$= D_{\max}^{(G)} - 2\sigma_{\hat{X}}\sigma_{\hat{Y}}. \tag{62}$$

It can be verified that

$$R \geq I(X; \hat{X})$$
$$= \frac{1}{2}\log(2\pi e \sigma_X^2) - h(X|\hat{X})$$
$$\geq \frac{1}{2}\log(2\pi e \sigma_X^2) - h(X - \hat{X})$$

$$\geq \frac{1}{2}\log(2\pi e\sigma_X^2) - \frac{1}{2}\log(2\pi e\mathbb{E}[(X-\hat{X})^2])$$

$$= \frac{1}{2}\log\frac{\sigma_X^2}{\sigma_X^2 - \sigma_{\hat{X}}^2},$$

which implies

$$\sigma_{\hat{X}} \leq \sigma_X\sqrt{1-2^{-2R}}. \tag{63}$$

Similarly,

$$\sigma_{\hat{Y}} \leq \sigma_Y\sqrt{1-2^{-2R}}. \tag{64}$$

Substituting (63) and (64) into (62) and invoking (55) in Theorem 6 shows

$$D_{\text{ncr}}^{(\infty)}(\mathcal{N}(\mu_X,\sigma_X^2),\mathcal{N}(\mu_Y,\sigma_Y^2),R) \geq D_{\min}^{(G)} + 2\sigma_X\sigma_Y 2^{-2R}.$$

To see that this lower bound is tight, we can let

$$X = \hat{X} + N, \tag{65}$$

$$Y = \hat{Y} + \hat{N}, \tag{66}$$

where $\hat{X} \sim \mathcal{N}(\mu_X,\sigma_X^2(1-2^{-2R}))$ is independent of $N \sim \mathcal{N}(0,\sigma_X^2 2^{-2R})$ while $\hat{Y} \sim \mathcal{N}(\mu_Y,\sigma_Y^2(1-2^{-2R}))$ is independent of $\hat{N} \sim \mathcal{N}(0,\sigma_Y^2 2^{-2R})$. This completes the proof of (58).

To prove (59) and (60), it suffices to note the well-known fact that $p_{\hat{X}|X}$ and $p_{\hat{Y}|Y}$ associated with (65) and (66) attain $D_{\text{mse}}^{(\infty)}(\mathcal{N}(\mu_X,\sigma_X^2),R)$ and $D_{\text{mse}}^{(\infty)}(\mathcal{N}(\mu_Y,\sigma_Y^2),R)$, respectively.

Now we proceed to prove (61). Consider $p_{X,Y} \in \Gamma(\mathcal{N}(\mu_X,\sigma_X^2),\mathcal{N}(\mu_Y,\sigma_Y^2))$ with $I(X;Y) \leq R$. Let $\xi \triangleq \mathbb{E}[(X-\mu_X)(Y-\mu_Y)]$. We have

$$\mathbb{E}[(X-Y)^2] = D_{\max}^{(G)} - 2\xi. \tag{67}$$

Moreover,

$$R \geq I(X;Y)$$
$$= \frac{1}{2}\log(2\pi e\sigma_X^2) + \frac{1}{2}\log(2\pi e\sigma_Y^2) - h(X,Y)$$
$$\geq \frac{1}{2}\log(2\pi e\sigma_X^2) + \frac{1}{2}\log(2\pi e\sigma_Y^2) - \frac{1}{2}\log((2\pi e)^2(\sigma_X^2\sigma_Y^2 - \xi))$$
$$= \frac{1}{2}\log\frac{\sigma_X^2\sigma_Y^2}{\sigma_X^2\sigma_Y^2 - \xi^2},$$

which implies

$$\xi \leq \sigma_X\sigma_Y\sqrt{1-2^{-2R}}. \tag{68}$$

Substituting (68) into (67) and invoking Theorem 7 shows

$$D_{\text{cr}}^{(\infty)}(\mathcal{N}(\mu_X,\sigma_X^2),\mathcal{N}(\mu_Y,\sigma_Y^2),R) \geq D_{\max}^{(G)} - 2\sigma_X\sigma_Y\sqrt{1-2^{-2R}}.$$

To see that this lower bound is tight, we can let $X$ and $Y$ be jointly Gaussian with $\xi = \sigma_X\sigma_Y\sqrt{1-2^{-2R}}$. This completes the proof of Theorem 8. We acknowledge that the expression of $D_{\text{cr}}^{(\infty)}(\mathcal{N}(\mu_X,\sigma_X^2),\mathcal{N}(\mu_Y,\sigma_Y^2),R)$ was established in (Saldi et al., 2015b, Section IV-B) for the special case when the Gaussian distributions have zero mean. However, to the best of our understanding, they only provided an upper bound for the case of no common randomness . □

In Figure 8, We plot $D_{\text{ncr}}^{(\infty)}(\mathcal{N}(\mu_X,\sigma_X^2),\mathcal{N}(\mu_Y,\sigma_Y^2),R)$, $\overline{D}_{\text{ncr}}^{(\infty)}(\mathcal{N}(\mu_X,\sigma_X^2),\mathcal{N}(\mu_Y,\sigma_Y^2),R)$, $\underline{D}_{\text{ncr}}^{(\infty)}(\mathcal{N}(\mu_X,\sigma_X^2),\mathcal{N}(\mu_Y,\sigma_Y^2),R)$, and $D_{\text{cr}}^{(\infty)}(\mathcal{N}(\mu_X,\sigma_X^2),\mathcal{N}(\mu_Y,\sigma_Y^2),R)$ for two illustrative examples.

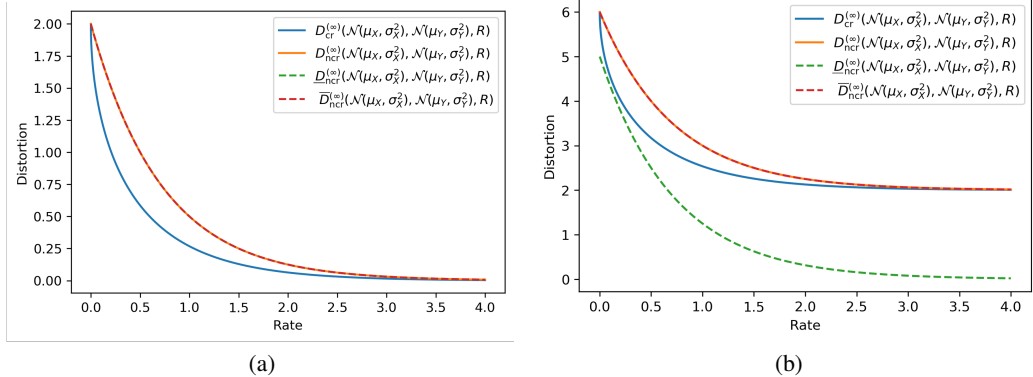

Figure 8: Gaussian case distortion-rate tradeoffs. (a) $\mu_X = \mu_Y = 0$, $\sigma_X = \sigma_Y = 1$, where $\overline{D}_{\mathrm{ncr}}^{(\infty)}(\mathcal{N}(\mu_X, \sigma_X^2), \mathcal{N}(\mu_Y, \sigma_Y^2), R)$ and $\underline{D}_{\mathrm{ncr}}^{(\infty)}(\mathcal{N}(\mu_X, \sigma_X^2), \mathcal{N}(\mu_Y, \sigma_Y^2), R)$ coincide with $D_{\mathrm{ncr}}^{(\infty)}(\mathcal{N}(\mu_X, \sigma_X^2), \mathcal{N}(\mu_Y, \sigma_Y^2), R)$; (b) $\mu_X = 0, \sigma_X = 1$, $\mu_Y = 1$, $\sigma_Y = 2$, where $\overline{D}_{\mathrm{ncr}}^{(\infty)}(\mathcal{N}(\mu_X, \sigma_X^2), \mathcal{N}(\mu_Y, \sigma_Y^2), R)$ is tight but $\underline{D}_{\mathrm{ncr}}^{(\infty)}(\mathcal{N}(\mu_X, \sigma_X^2), \mathcal{N}(\mu_Y, \sigma_Y^2), R)$ is loose. Moreover, it can be seen from both examples that common randomness can indeed help improve the distortion-rate tradeoff.

## B  EXPERIMENTAL RESULTS

### B.1  DATASET

Image super-resolution is conducted on MNIST (LeCun et al., 1998). For synthesizing low-resolution images, we perform bilinear downsampling on the original image from $28 \times 28$ to $7 \times 7$. The samples in Figure 5(b) show that the low resolution digits are blurry and some of them are hard to recognize. Image denoising is conducted on SVHN (Netzer et al., 2011). In our experiments, we synthesize the noisy image with additive Gaussian noise. The standard deviation is set to 20.

### B.2  UNIVERSAL QUANTIZATION

Let $\mathcal{C}$ be our codebook for quantization. Recall that the encoder uses a $\tanh$ activation so its output lies in $(-1, 1)^d$. Given dimension $d$ and $L$ quantization levels per dimension as parameters, $\mathcal{C}$ will consist of $L$ uniformly spaced intervals across all $d$ dimensions. The upper bound of model rate is given by $d \log(L)$. With this codebook, universal quantization (Ziv, 1985; Gray & Stockham, 1993; Theis & Agustsson, 2021) is implemented as follows. We assume the sender and receiver have access to the same $u \sim U[-1/(L-1), +1/(L-1)]^d$. The sender computes

$$z = \arg\min_{c \in \mathcal{C}} \|f(x) + u - c\|$$

and sends $z$ to the receiver. The receiver decodes the image by passing $z - u$ through the decoder. This is also known as a subtractive dither in literature (Gray & Stockham, 1993). For super-resolution and image denoising, the interval $L$ is respectively fixed at 4 and 8.

### B.3  TRAINING DETAILS

To induce distributional shift, we use the Wasserstein GAN for our experiments. We alternate between training the encoder/decoder $f, g$, and the critic $h$. By Kantorovich-Rubinstein duality (Villani, 2009), the critic is used to approximate

$$W_1(p_Y, p_{\tilde{Y}}) = \sup_{\|\nabla h\| \leq 1} \mathbb{E}[h(Y)] - \mathbb{E}[h(\tilde{Y})], \tag{69}$$

where $\tilde{Y} = g(Q(f(X)+U)-U)$ for $U$ as in Appendix B.2. The Lipschitz constraint is implemented with a gradient penalty (Gulrajani et al., 2017) in practice.

**Super-resolution.** The training for end-to-end network lasts for 50 epochs. $\lambda$ in (15) is fixed at $1e-3$ across all rates. The learning rate initialized to be 0.0001 and is decayed by a factor of 5 after 30 epochs. The Adam (Kingma & Ba, 2014) optimizer is used. Table 3 illustrates the detailed training setting. For the helper two-branch network at a specific rate constraint, we load the pre-trained encoder weight of the corresponding end-to-end network, as well as two randomly initialized decoders $g_1, g_2$. Note that only theese two decoders are trainable. During training, we use the Adam optimizer with the learning rate initialized at 0.0001. There are a total of 100 epochs until the convergence of the two decoders. The learning rate is decayed once at 50 epochs by a factor of 5. Detailed training settings are shown in Table 4.

**Image Denoising.** The experiments for image denoising share many settings with image super-resolution. Tables 3 and 4 can be reused to reproduce the experiments on image denoising. Here, we list the difference between them. For denoising, the end-to-end model is trained for 100 epochs with $\lambda$ fixed at $3e-3$ across all rates. The learning rate is decayed by a factor of 5 after 40 epochs. The two-branch model is trained for total 200 epochs and we decay the learning rate at 100 epochs by a factor of 5.

Table 3: Hyperparameters used for training end-to-end model in Fig. 4(a). $\alpha$ is the learning rate, $(\beta_1, \beta_2)$ are the parameters for Adam, and $\lambda_{\mathrm{GP}}$ is the gradient penalty coefficient.

|          | $\alpha$   | $\beta_1$ | $\beta_2$ | $\lambda_{\mathrm{GP}}$ |
|----------|------------|-----------|-----------|------------|
| Encoder  | $10^{-4}$  | 0.5       | 0.999     | -          |
| Decoder  | $10^{-4}$  | 0.5       | 0.999     | -          |
| Critic   | $10^{-4}$  | 0.5       | 0.999     | 10         |

Table 4: Hyperparameters used for training two-branch model in Fig. 4(b). $\alpha$ is the learning rate, $(\beta_1, \beta_2)$ are the parameters for Adam, and $\lambda_{\mathrm{GP}}$ is the gradient penalty coefficient.

|           | $\alpha$   | $\beta_1$ | $\beta_2$ | $\lambda_{\mathrm{GP}}$ |
|-----------|------------|-----------|-----------|------------|
| Encoder   | 0          | -         | -         | -          |
| Decoder-1 | $10^{-4}$  | 0.5       | 0.999     | -          |
| Decoder-2 | $10^{-4}$  | 0.5       | 0.999     | -          |
| Critic    | $10^{-4}$  | 0.5       | 0.999     | 10         |

### B.4 DETAILED RESULTS IN FIGURE. 5

In Fig. 5(a)(c), we have provided a comparison between the case with or without common randomness in the form of a scatter chart. Here, we present detailed quantities of each point in Fig. 5(a)(c). Table 5 shows the number of each dot for super-resolution experiments, and Table 6 present the value of each dot for image denoising. From the two tables, we can further see the utility of common randomness quantitatively.

### B.5 COMPARISON WITH BASELINE

To illustrate the effectiveness of our system, we compare with a baseline method that separately deal with the tasks of image restoration and compression. For the restoration (image super-resolution and denoising), we respectively build two U-Nets with skip connections and train them in the unsupervised manner by adopting the Eq. 15 as objective i.e.,

$$L_1 = \mathbb{E}[\|X - \tilde{Y}\|^2] + \lambda_1 W_1(p_Y, p_{\tilde{Y}}). \tag{70}$$

After the restoration networks are trained to converge, we fix their weights and use them to produce restored images $\tilde{Y}$ given degraded one $X$. Afterwards, we adopt our end-to-end network as compression network by minimizing the following loss at different rates:

Table 5: The detailed number of MSE distortion loss in Fig. 5(a).

Super-resolution **with** Common Randomness

| Rate | 4 | 6 | 8 | 10 | 12 | 14 | 16 | 18 |
|------|------|------|------|------|------|------|------|------|
| MSE | 0.0515 | 0.0457 | 0.0420 | 0.0394 | 0.0372 | 0.0353 | 0.0339 | 0.0324 |
| Rate | 20 | 22 | 24 | 26 | 28 | 30 | 32 | - |
| MSE | 0.0313 | 0.0300 | 0.0297 | 0.0285 | 0.0280 | 0.0277 | 0.0269 | - |

Super-resolution **without** Common Randomness

| Rate | 4 | 6 | 8 | 10 | 12 | 14 | 16 | 18 |
|------|------|------|------|------|------|------|------|------|
| MSE | 0.0558 | 0.0506 | 0.0463 | 0.0435 | 0.0415 | 0.0396 | 0.0383 | 0.0367 |
| Rate | 20 | 22 | 24 | 26 | 28 | 30 | 32 | - |
| MSE | 0.0351 | 0.0337 | 0.0331 | 0.0328 | 0.0315 | 0.0308 | 0.0300 | - |

Table 6: The detailed number of MSE distortion loss in Fig. 5(c).

Image Denoising **with** Common Randomness

| Rate | 12 | 18 | 24 | 30 | 36 | 42 | 48 | 54 | 60 |
|------|------|------|------|------|------|------|------|------|------|
| MSE | 0.0219 | 0.0195 | 0.0175 | 0.01634 | 0.0154 | 0.0147 | 0.0138 | 0.0135 | 0.0129 |
| Rate | 66 | 72 | 78 | 84 | 90 | 96 | 102 | 108 | 114 |
| MSE | 0.0126 | 0.0123 | 0.0118 | 0.0117 | 0.0115 | 0.0112 | 0.0109 | 0.0107 | 0.0106 |

Image Denoising **without** Common Randomness

| Rate | 12 | 18 | 24 | 30 | 36 | 42 | 48 | 54 | 60 |
|------|------|------|------|------|------|------|------|------|------|
| MSE | 0.0230 | 0.0208 | 0.0189 | 0.0175 | 0.0165 | 0.0157 | 0.0151 | 0.0145 | 0.014 |
| Rate | 66 | 72 | 78 | 84 | 90 | 96 | 102 | 108 | 114 |
| MSE | 0.0137 | 0.0133 | 0.0130 | 0.0127 | 0.0123 | 0.0120 | 0.0118 | 0.0116 | 0.0114 |

$$L_1 = \mathbb{E}[\|\tilde{Y} - Y^+\|^2] + \lambda W_1(p_Y, p_{Y+}), \tag{71}$$

where $Y^+$ is the outputs of compression network. Note that, to guarantee the distribution of reconstructed $Y^+$ is close to that of target images, we implement a penalty on the Wasserstein-1 distance in (70) and (71). For the image super-resolution, we experimentally selected $\lambda_1 = 0.05$ and $\lambda_2 = 0.01$. For the image denoising, we experimentally selected $\lambda_1 = 0.03$ and $\lambda_2 = 0.005$. Once the compression network is converged, we report the final MSE distortion between $Y^+$ and $X$ using $\mathbb{E}[\|X - Y^+\|^2]$.

The detailed results for entropy-constrained image super-resolution and denoising are respectively shown in Table 7 and Table 8. It can easily check throughout the tables that our end-to-end systems outperform the baselines.

## B.6 COMPARISON WITH GROUND TRUTH

In order to illustrate the rate-distortion trade-offs, we report the MSE distortion that is measured between *degraded input images* and decoder outputs in Figure 5. Since the input and output distributions are different, we do not expect MSE $\rightarrow 0$ as the rate increases. The MSE distortion between degraded input and restored output is still able to reveal how much content information of the input is preserved in output (lower is better).

We now additionally show the MSE distortion between *ground truth* and decoder outputs in Tables 9 and 10. Concretely, we measure MSE distortion $\mathbb{E}[\|Y - \tilde{Y}\|^2]$, where $Y$ is ground truth and $\tilde{Y}$ is the network output. Note that *for training, the ground truth is only used in an unsupervised fashion*

Table 7: Comparison between our end-to-end system with the baseline method for image super-resolution. Numbers are the MSE distortion loss for a particular rate. Best results are in **bold**.

Super-resolution **with** Common Randomness

| Rate | 4 | 6 | 8 | 10 | 12 | 14 | 16 | 18 |
|---|---|---|---|---|---|---|---|---|
| Baseline | 0.0603 | 0.0568 | 0.0544 | 0.0530 | 0.0523 | 0.0511 | 0.0503 | 0.0498 |
| Ours | **0.0515** | **0.0457** | **0.0420** | **0.0394** | **0.0372** | **0.0353** | **0.0339** | **0.0342** |
| Rate | 20 | 22 | 24 | 26 | 28 | 30 | 32 | - |
| Baseline | 0.0489 | 0.0485 | 0.0484 | 0.0482 | 0.0478 | 0.0476 | 0.0471 | - |
| Ours | **0.0313** | **0.0300** | **0.0297** | **0.0285** | **0.0280** | **0.0277** | **0.0269** | - |

Super-resolution **without** Common Randomness

| Rate | 4 | 6 | 8 | 10 | 12 | 14 | 16 | 18 |
|---|---|---|---|---|---|---|---|---|
| Baseline | 0.0620 | 0.0585 | 0.0573 | 0.0555 | 0.0543 | 0.0535 | 0.0532 | 0.0520 |
| Ours | **0.0558** | **0.0506** | **0.0463** | **0.0435** | **0.0415** | **0.0396** | **0.0383** | **0.0367** |
| Rate | 20 | 22 | 24 | 26 | 28 | 30 | 32 | - |
| Baseline | 0.0517 | 0.0512 | 0.0506 | 0.0505 | 0.0497 | 0.0495 | 0.0490 | - |
| Ours | **0.0351** | **0.0337** | **0.0331** | **0.0328** | **0.0315** | **0.0308** | **0.0300** | - |

Table 8: Comparison between our end-to-end system with the baseline method for image denoising. Numbers are the MSE distortion loss for a particular rate. Best results are in **bold**.

Image Denoising **with** Common Randomness

| Rate | 12 | 18 | 24 | 30 | 36 | 42 | 48 | 54 | 60 |
|---|---|---|---|---|---|---|---|---|---|
| Baseline | 0.0242 | 0.0213 | 0.0189 | 0.0173 | 0.0162 | 0.0154 | 0.0148 | 0.0142 | 0.0136 |
| Ours | 0.0219 | 0.0195 | 0.0175 | 0.0163 | 0.0154 | 0.0147 | 0.0138 | 0.0135 | 0.0129 |
| Rate | 66 | 72 | 78 | 84 | 90 | 96 | 102 | 108 | 114 |
| Baseline | 0.0134 | 0.0130 | 0.0127 | 0.0124 | 0.0120 | 0.0118 | 0.0116 | 0.0111 | 0.0110 |
| Ours | 0.0126 | 0.0123 | 0.0118 | 0.0117 | 0.0115 | 0.0112 | 0.0109 | 0.0107 | 0.0106 |

Image Denoising **without** Common Randomness

| Rate | 12 | 18 | 24 | 30 | 36 | 42 | 48 | 54 | 60 |
|---|---|---|---|---|---|---|---|---|---|
| Baseline | 0.0252 | 0.0216 | 0.0202 | 0.0185 | 0.0178 | 0.0164 | 0.0158 | 0.0154 | 0.0149 |
| Ours | 0.0230 | 0.0208 | 0.0189 | 0.0175 | 0.0165 | 0.0157 | 0.0151 | 0.0145 | 0.014 |
| Rate | 66 | 72 | 78 | 84 | 90 | 96 | 102 | 108 | 114 |
| Baseline | 0.0147 | 0.0144 | 0.0140 | 0.0135 | 0.0133 | 0.0129 | 0.0126 | 0.0120 | 0.0117 |
| Ours | 0.0137 | 0.0133 | 0.0130 | 0.0127 | 0.0123 | 0.0120 | 0.0118 | 0.0116 | 0.0114 |

*with unpaired noisy images*, and here the ground truth-noisy image pairs are only used for test time evaluation. Note also that the MSE distortion is correspondingly lower if common randomness is adopted.

Table 9: Illustration of MSE distortion between network outputs and ground truth for super-resolution.

Super-resolution **with** Common Randomness

| Rate | 4 | 8 | 12 | 16 | 20 | 24 | 28 | 32 |
|---|---|---|---|---|---|---|---|---|
| MSE | 0.0582 | 0.0464 | 0.0408 | 0.0380 | 0.0360 | 0.0353 | 0.0348 | 0.0343 |

Super-resolution **without** Common Randomness

| Rate | 4 | 8 | 12 | 16 | 20 | 24 | 28 | 32 |
|---|---|---|---|---|---|---|---|---|
| MSE | 0.0646 | 0.0492 | 0.0426 | 0.0390 | 0.0375 | 0.0362 | 0.0353 | 0.0345 |

Table 10: Illustration of MSE distortion between network outputs and ground truth for image denoising.

### Image Denoising **with** Common Randomness

| Rate | 12 | 24 | 36 | 48 | 60 | 72 | 84 | 96 | 108 |
|------|------|------|------|------|------|------|------|------|------|
| MSE | 0.0157 | 0.0114 | 0.0092 | 0.0077 | 0.0069 | 0.0062 | 0.0056 | 0.0052 | 0.0047 |

### Image Denoising **without** Common Randomness

| Rate | 12 | 24 | 36 | 48 | 60 | 72 | 84 | 96 | 108 |
|------|------|------|------|------|------|------|------|------|------|
| MSE | 0.0169 | 0.0128 | 0.0104 | 0.0090 | 0.0080 | 0.0072 | 0.0066 | 0.0060 | 0.0057 |

## B.7    BREAKDOWN OF THE TABLE 1

It is worth reminding that the each number in Table 1 is the total distortion 18. Here, we provide a breakdown of total distortion in each of the three term for joint training, i.e, $\mathbb{E}[\|X - \tilde{Y}_1\|^2]$, $\mathbb{E}[\|\tilde{Y} - \tilde{Y}_2\|^2]$ and $\mathbb{E}[\|\tilde{Y}_1 - \tilde{Y}_2\|^2]$.

## B.8    NETWORK ARCHITECTURE

**Super-resolution**. The detailed network structure for end-to-end model and two-branch model are respectively presented in Table 12 and Table 13. The last linear layer of the encoder controls the number of output symbols.

**Denoising**. The detailed network structure for end-to-end model and two-branch model are respectively presented in Table 14 and Table 15. The last linear layer of the encoder controls the number of output symbols.

Table 11: Breakdown of the Table 1. At any rate, it can be observed that the total losses of our approximation system ($L_2$) are very close to that of end-to-end learning system under the setting without common randomness ($\mathbb{E}[\|X - \tilde{Y}\|^2]$).

### Image Super-resolution Using Joint Training

| Rate | $\mathbb{E}[\|X - \tilde{Y}_1\|^2]$ | $\mathbb{E}[\|\tilde{Y} - \tilde{Y}_2\|^2]$ | $\mathbb{E}[\|\tilde{Y}_1 - \tilde{Y}_2\|^2]$ | $L_2$ | $\mathbb{E}[\|X - \tilde{Y}\|^2]$ |
|------|------|------|------|------|------|
| 4 | 0.0355 | 0.0227 | 0.0004 | **0.0586** | **0.0558** |
| 10 | 0.0223 | 0.0222 | 0.0008 | **0.0453** | **0.0435** |
| 20 | 0.0136 | 0.0191 | 0.00013 | **0.0349** | **0.0351** |
| 30 | 0.00122 | 0.0172 | 0.0015 | **0.0309** | **0.0308** |

### Image Denoising Using Joint Training

| Rate | $\mathbb{E}[\|X - \tilde{Y}_1\|^2]$ | $\mathbb{E}[\|\tilde{Y} - \tilde{Y}_2\|^2]$ | $\mathbb{E}[\|\tilde{Y}_1 - \tilde{Y}_2\|^2]$ | $L_2$ | $\mathbb{E}[\|X - \tilde{Y}\|^2]$ |
|------|------|------|------|------|------|
| 12 | 0.01911 | 0.00497 | 0.00020 | **0.02428** | **0.02302** |
| 30 | 0.01458 | 0.00435 | 0.00026 | **0.01919** | **0.01746** |
| 60 | 0.01168 | 0.00378 | 0.00032 | **0.01578** | **0.01401** |
| 90 | 0.01035 | 0.00323 | 0.00028 | **0.01386** | **0.01229** |

Table 12: Model architectures of end-to-end network used in super-resolution.

| Encoder |
|---|
| Input |
| Conv2D, l-ReLU |
| Conv2D, l-ReLU |
| Flatten |
| Linear, l-ReLU |
| Linear, l-ReLU |
| Linear, Tanh |
| Quantizer |

| Decoder |
|---|
| Input |
| Linear, BatchNorm1D, l-ReLU |
| Linear, BatchNorm1D, l-ReLU |
| Unflatten |
| ConvT2D, BatchNorm2D, l-ReLU |
| ConvT2D, BatchNorm2D, l-ReLU |
| ConvT2D, BatchNorm2D, Sigmoid |

| Critic |
|---|
| Input |
| Conv2D, l-ReLU |
| Conv2D, l-ReLU |
| Conv2D, l-ReLU |
| Linear |

Table 13: Model architectures of two-branch network used in super-resolution.

| Encoder |
|---|
| Input |
| Conv2D, l-ReLU |
| Conv2D, l-ReLU |
| Flatten |
| Linear, l-ReLU |
| Linear, l-ReLU |
| Linear, Tanh |
| Quantizer |

| Decoder1 and 2 |
|---|
| Input |
| Linear, BatchNorm1D, l-ReLU |
| Linear, BatchNorm1D, l-ReLU |
| Unflatten |
| ConvT2D, BatchNorm2D, l-ReLU |
| ConvT2D, BatchNorm2D, l-ReLU |
| ConvT2D, Sigmoid |

Table 14: Model architectures of end-to-end network used in image denoising.

| Encoder |
|---|
| Input |
| Conv2D, l-ReLU |
| Conv2D, l-ReLU |
| Conv2D, l-ReLU |
| Flatten |
| Linear, Tanh |
| Quantizer |

| Decoder |
|---|
| Input |
| Linear, BatchNorm1D, l-ReLU |
| Linear, BatchNorm1D, l-ReLU |
| Unflatten |
| ConvT2D, BatchNorm2D, l-ReLU |
| ConvT2D, BatchNorm2D, l-ReLU |
| ConvT2D, BatchNorm2D, l-ReLU |
| ConvT2D, BatchNorm2D, Sigmoid |

| Critic |
|---|
| Input |
| Conv2D, l-ReLU |
| Conv2D, l-ReLU |
| Conv2D, l-ReLU |
| Linear |

Table 15: Model architectures of two-branch network used in image denoising. ResBlock is formed using two Conv2D and skip connection.

| Encoder |
|---|
| Input |
| Conv2D, l-ReLU |
| Conv2D, l-ReLU |
| Conv2D, l-ReLU |
| Flatten |
| Linear, Tanh |
| Quantizer |

| Decoder1 and 2 |
|---|
| Input |
| Linear, BatchNorm1D, l-ReLU |
| Linear, BatchNorm1D, l-ReLU |
| Unflatten |
| ConvT2D, l-ReLU |
| ResBlock, ConvT2D, l-ReLU |
| ResBlock, ConvT2D, l-ReLU |
| ResBlock, ConvT2D, Sigmoid |

