# OpenReview forum: "LOSSY COMPRESSION WITH DISTRIBUTION SHIFT AS ENTROPY CONSTRAINED OPTIMAL TRANSPORT"
_ICLR.cc/2022/Conference — ICLR 2022 Poster_

### Official Review · Reviewer_LfvG · 2021-10-26

**Correctness:** 1
**Technical Novelty And Significance:** 1
**Empirical Novelty And Significance:** 1
**Recommendation:** 3
**Confidence:** 4

**Main Review:**

Strength: Extend the conventional concept of the optimal transport to handle perception (constraint on the output probability distribution).

Weakness:
(1) Relevance of the studied quantities to practical data compression is unclear.
(2) Theorems 1 and 3 seem logically inconsistent with each other.

Detail of (1): in the footnote of page 3 says that any DISCRETE RV Z can be losslessly compressed to at most H(Z)+1 bits. The point
is Z being DISCRETE. with finite-support. In order for Definitions 2 and 3 to have practical significance, the RVs to be compressed must
be discrete and finite-support, in particular, Z in Definitions 2 and 3 must be finite-support.
The problem formulations do not seem relevant for practical data compression,

Detail of (2): In the problem formulation (Definition 3) and its solution (Theorem 3), there is no condition on the common randomness U.
Taking H(U)=0, i.e., U=0 always should reduce Definition 3 to Definition 2 and Theorem 3 to Theorem 2, but it does not.
There seems to be some unstated minimally required entropy on U, i.e., H(U) must be larger than some unstated constant number.
For example, page 5 states "This is in contrast to the case without common randomness in Theorem 1 where the reconstruction Y
must be generated at the decoder." Is this true even with U of H(U) = 1 / 1,000,000,000??

**Summary Of The Paper:**

* Characterize the optimal transport under the distortion and the output probability distribution constraints.
* Provide DNN methods to evaluate the above quantities.

**Summary Of The Review:**

The relevance of the paper to practical data compression is unclear, and it seems that mathematical quantities without practical relevance
are considered. The stated theorems look false.

---

> ### Author Response · Authors · 2021-11-18
> **Author Response**
>
> 1. *Relevance to Practical Data Compression*: In our problem formulation, the input $X$ (i.e., the source to be compressed) and the reconstruction $Y$  are not necessarily discrete as noted in Section $2$.  Please note that the source $X$ is mapped to a representation $Z$  at the encoder via a conditional distribution  $p_{Z|X}$ which is under our control. We naturally let $Z$  be discrete so that we can represent it using a finite number of bits.   We believe that our formulation is standard and compatible with real compression problems.
>
> 2. *Constraints on the random variable $U$*: In Definition 3 and in Theorem 3 we optimize over all possible distributions $P_U$ (please see the comment following Eq. (11) in the revised version). This is the reason why the characterization of the minimum achievable distortion in Theorem 3 does not depend on the distribution $P_U$ or the entropy $H(U)$. In other words, the result in Theorem 1 cannot be extracted as a special case of Theorem 3 and there is no inconsistency between them. We have noted in the Conclusions that extending our formulation to impose additional constraints on the shared randomness could be an interesting potential research direction.

---

> > ### Comment · Reviewer_LfvG · 2021-11-18
> > **I was unable to see how practical the research is.**
> >
> > Thank you very much for your response.
> >
> > > Please note that the source X is mapped to a representation Z at the encoder via a conditional distribution p_{Z|X} which is under our control.
> >
> > My concern was that, Definition 2 (or Definition 1) did not ensure the existence of an encoder.
> > The concern still remains in the revised manuscript.
> >
> > > We believe that our formulation is standard and compatible with real compression problems.
> >
> > I disagree. When the optimal (smallest) compression rate is considered in the information theory,
> > the problem formulation explicitly requires existence of encoder (and decoder). In that sense,
> > I do not see Definition 2 (or Definition 1) as a standard one.
> > If the quantity in Definition 2 is not related to existence of encoder, then its practical relevance is unclear at best.
> >
> > > In Definition 3 and in Theorem 3 we optimize over all possible distributions P_U (please see the comment following Eq. (11) in the revised version).
> >
> > This means that H(U) can be arbitrary large. Very large H(U) means that extremely large amount of randomness
> > is possibly required to be shared between encoder and decoder. I do not think such assumption is practical.
> >
> > My overall impression remains the same as my first review comment: The practical relevance of the studied
> > mathematical quantities are unclear at best.

---

> > > ### Author Response · Authors · 2021-11-18
> > > **Followup on Practical Relevance**
> > >
> > >
> > >
> > > **Existence of Encoder and Decoder:**
> > >
> > > We would like to clarify that any choice of $p_{Z|X}$ and $p_{Y|Z}$ (along with lossless entropy coding) by themselves form an encoder-decoder pair. No additional argument is needed to prove the existence.  We elaborate this further below.
> > >
> > > In the standard information theory formulation, there is an encoder that maps the source $X$ to a bit sequence of length $R$ and a decoder that maps the bit sequence to the reconstruction $Y$. Our formulation is compatible with this standard formulation in the sense that $p_{Z|X}$ together with an entropy encoder for $Z$ yields a mapping from $X$ to a bit sequence of length approximately $H(Z)$ while the entropy decoder for $Z$ along with $p_{Y|Z}$ yields a mapping from the bit sequence to $Y$.
> > > This is also the formulation adopted in the following paper:
> > >
> > > *Yan et al., On Perceptual Lossy Compression: The Cost of Perceptual Reconstruction and An Optimal Training Framework, ICML 2021*
> > >
> > >
> > > The only difference with classical information theory problems (e.g., channel coding and lossy source coding), is that in these settings the encoder and decoder are often assumed to be deterministic while here they are allowed to be stochastic. Our formulation is basically modelled after that of entropy-constrained quantization (Gray and Neuhoff, "Quantization," *IEEE Transactions on Information Theory*, 1998) with additional stochasticity known to be necessary for addressing the output distribution constraint motivated by perception considerations. As an aside, please note that $p_{Z|X}$ should not be associated with a test channel in the proof of the classic rate-distortion theorem.
> > >
> > >
> > >
> > >
> > >
> > >
> > > **Entropy of $U$:**
> > >
> > > Definition 3 and Theorem 3 intend to characterize the maximum benefit that can be offered by shared randomness in a theoretical setting.   Leveraging shared randomness has a long history in both the theory and practice of data compression  (as evidenced by a vast body of literature on dithered quantization, see, e.g., Gray and Stockham, "Dithered Quantizers," *IEEE Transactions on Information Theory*, 1993) and more recently finds applications in learning-related lossy compression as noted in our submission.   Indeed our experiments also demonstrate that such schemes that use shared randomness can already provide noticeable gains in the rate-distortion performance.
> > >
> > > Practical implementations for shared randomness  are also common.  For example, time-based common randomness generation is routinely used in practice (see, e.g., Time-based One-time Password (TOTP)). Many pseudo-random number generators are initialized based on the current system clock (see, e.g., *srandom( )* and *random*( ) from C’s standard library) and shared randomness can be extracted when systems are synchronized.  Motivated by such considerations, we do not impose a constraint on the entropy of $U$, but optimize the distribution of $U$ and the design of $p_{Z|X,U}$ and $p_{Y|Z,U}$ in Definition 3, which is of practical relevance.

---

> ### Author Response · Authors · 2021-11-24
> **Follow up**
>
> We wanted to send a gentle reminder since we have not seen a follow up to our response (on Nov 18th) to the questions you had asked then. We hope to have addressed all your concerns satisfactorily. If you have no further concerns we would appreciate such a clarification. We also kindly request you to consider increasing the score.

---

### Official Review · Reviewer_GKce · 2021-10-31

**Correctness:** 4
**Technical Novelty And Significance:** 3
**Empirical Novelty And Significance:** 3
**Recommendation:** 8
**Confidence:** 3

**Main Review:**

The authors consider a scenario where an encoder observes some signal X and transmits it to a decoder as a quantized version ^X. At the decoder, this signal is transported into another signal called ^Y, from which the reconstruction Y is computed. The key point is that X and Y are not distributed wrt the same distributions: basically, X~Px is a degraded version of Y~Py.

In this setup, the very nice contribution of the paper is to express the whole problem as a particular instance of optimal transport, that indeed appears as the right framework to modify X~Py so that it becomes representative of Py. With this in mind, the "compression" part comes into the picture as the introduction of a latent variable ^X with limited entropy.

Having some personal background in source coding, I personally found the whole story very appealing and the derivations look very well grounded. Although of course the introduction of optimal transport in the picture reads exotic to my eyes, it sounds perfectly natural and everything is provided to allow the reader to understand (at least almost).

The experiment look rather weak in my opinion, and I would definitely have liked a much stronger support for the theory, possibly showing me the applicability of the method on real-life problems/scales. Still, the few experiments that are provided are ok and look promising already.

Before providing my detailed feedback, I must hence say that I recommend acceptance of this paper.

Now, I have a few questions/remarks that I will list here in no particular order:
* "restoration at once using" simultaneously
* "lower than the case when" lower than when
* what is it exactly you are calling "geometrically representative of X and Y" ? This doesn't make sense to me
* are you sure you want to refer to (2) when you write "The entropy constraint H(^X)<=R in (2) essentially guarantees that"
* When you write "Finally, it is worth noting that under mild regularity conditions, the optimal converter can be realized as a (deterministic) bijection)", it looks to me you are having the scalar case in mind, which indeed looks easy as an increasing rearrangement. However, how would you proceed when your code ^X is multidimensional ? My understanding is that optimal transport in this case is not trivial at all. I see two main ways to circumvent the problem: 1. Applying transport independently in each dimension after assuming they are independent (are they?) 2. Assuming the dictionary for ^X is finite so that some histogram transporting method like sinkhorn etc could be used. Could you please be clearer on these points, or tell me if I'm mistaken somewhere ?
* I understand that it makes sense to assume that U and X are independent. However, I would be curious at what happens when they are not. It turns out I worked in such cases of "informed coding" where both coder and decoder were sharing information that is dependent on the signal to convey, and I kind of feel that the framework you propose could be compatible with this kind of ideas still. (Think for instance of the case where you want to transmit ^X, but where both the encoder and the decoder know some additional statistics that depend on X).
Is independence mandatory ?
* what do you mean exactly by "settiong U to be a constant" ? you wean that Pu is a dirac centered on some unique value ? I am asking because that sentence had me confused: do we agree that even if we assume it is random, we do assume that U is available at both coder and decoder (in the shared noise setting) ?
*  When you assume that H(Y|U,X)=0, are you basically assuming that if you have U and X, then you would perfectly know Y ? What is the meaning of this assumption ?
* "compress it lossless" losslessly
* I am sorry to say that section 2.2 is really not "reader friendly" enough for me and that I couldn't follow that one. I would really recommend putting some technical content to the appendix and rather spend space explaining what happens.
* It looks like you wanted to write (c) and not two times (b) in the caption of figure 3, right ?
* After theorem 3, you mention you need Y|U to be discrete. Is there no way to relax this to delve into the domains of differential entropy ? I would be curious about what you think would be the next steps in a direction that would relax this discretization assumption

* importantly, I can see that "in practice, we do not impose a hard equality constraint" on entropy. So actually, it looks to me that you are *not* implementing your "entropy constrained" OT, but rather some entropy-penalty. Although I understand it goes it the same direction, all your previous derivations seem to be violated by these practical choices. I guess it would be very important to at least hint at how this particular choice is under-optimal in the rate-distortion setups you developed ? I guess that this fact makes previous work about Sinkhorn distances more related to your work than explained in the paper, right ?

* could you please detail how h is trained exactly ? through adversarial training ? with a WGAN setup ?

* So, after all these developments, it looks like it is almost optimal already to just train separate encoders and decoders rather than a joint optimization ? Do you see scenarios where the derived theory would prove essential ?

**Summary Of The Paper:**

The paper focuses on a very particular kind of lossy compression, which they call "cross-domain lossy compression". It basically amounts to encode a degraded version of the signal at the encoder while being able to reconstruct some better version at the decoder, while maintaining interesting rate constraints. As such, this scenario puts together signal compression and denoising/super-resolution, so that I can definitely say it comes with interesting applications.

Technically, the authors show that the problem can be nicely formulated as a special kind of optimal transport under entropy constraints, which gives an interesting flavour to their contribution and much theoretical grounding for their proposed method.

**Summary Of The Review:**

original scenario for source coding that may be of interest to some specific applications. The connection with optimal transport is definitely interesting. There is much to do regarding validating the proposed approach in real large scale applications.

---

> ### Author Response · Authors · 2021-11-18
> **Author Response -- Part 1**
>
> We thank the reviewer for the detailed feedback and suggestions.
> 1. the typos in the first two comments have been fixed.
> 2. *geometrically representative of $X$ and $Y$*: We simply wanted to refer to *discretized* versions of $X$ and $Y$ like in Figure 1. This is fixed.
> 3. *Reference to Eq. (2)*: This has been fixed.
> 4. *comment regarding computation of optimal transport map*:  We thank the reviewer for the comment and note the following: 1)  we would like to kindly clarify that the claim on sufficiency of a deterministic bijection between $\hat{X}$ and $\hat{Y}$ does not restrict these variables to be scalar valued. Indeed Theorem 2 holds for the general case of multi-dimensional variables. We are sorry for any confusion our reference to Fig. 1 may have caused and  have updated our discussion.  2) The optimal transport map (associated with the Wasserstein distance) can be readily computed for the case of scalar variables.  We have  provided an  example involving continuous-valued uniformly distributed sources in Appendix A6 which uses such computations. 3) For the case of general multi-dimensional variables, the computation of optimal transport map in Theorem 1 may not be  straightforward.  In our experimental section this problem is tackled  using deep learning models.  We train end-to-end deep learning models which effectively learn mappings that approximate the optimal transport map,  and we verify the decomposition presented in Theorem 1 experimentally in Table 1.
>
>
>     We have added a remark along these lines in the paragraph after Theorem 2 in the paper.
>
> 5. *Dependence between $U$ and $X$*:  We agree with the reviewer that the case when variables $U$ and $X$ are correlated could have interesting practical applications. We have not pursued such an extension in the present work and have noted in the Conclusions section that this can be an interesting future direction.
>
>
> 6. *Randomness of $U$*: The reviewer correctly notes that in our setup $U$ is random and revealed to both the encoder and decoder. We optimize over all possible distributions $P_U$ in the characterization presented in Theorem 3. We are sorry if the sentence ``$U =$ constant" caused confusion and we have avoided it in the revised version.   Our only reason for bringing up this point was to argue that $D_{ncr} \ge D_{cr} $. We  have emphasized in the revised version that our setup and the associated characterization in Theorem 3 involves optimizing over all possible distributions $P_U$.
>
>
> 7. *Condition that $H(Y|U,X)=0$*:  In Theorem 3 the constraint $H(Y|U,X)=0$  (i.e., $Y$ is a deterministic function of $U$ and $X$) is imposed without loss of optimality (please see the proof in the Appendix). Thus even for this more restrictive family of distributions, we can achieve the same minimum distortion as in Definition 3. The advantage of the characterization in Theorem 3 is that it is amenable to evaluation for discrete alphabets as the problem of minimizing the distortion can be written in the form of a linear program, as noted in the Appendix A3.
> 8. *"compress it lossless" $\rightarrow$  losslessly*: fixed

---

> ### Author Response · Authors · 2021-11-18
> **Author Response --- Part 2**
>
> 9. *Section illustrating binary case*: the main takeaways are that (1) common randomness reduces the coding cost, and that (2) the upper bound can be tight, so that using optimal quantizer and dequantizer in the conventional  sense of minimizing distortion under a rate constraint incurs no penalty.  We have rewritten the section as suggested by the reviewer.
>
> 10. *Fig. 3 Caption (c)*: thanks for pointing this out. We have corrected it.
>
>  11. *Use of Differential Entropy in Theorem 3*:  Unfortunately we do not believe it is straightforward to relax the assumption that the conditional distribution $Y|U$ is discrete. If $Y|U$ is continuous and even if the differential entropy is finite we will need an infinite number of bits for representing the output. We note that the requirement that the distribution $Y|U$ be discrete also appears in other problems involving one-shot rate-distortion in information theory (e.g. Li and El Gamal (2018)).
>
> 12. *concern on "in practice, we do not impose a hard equality constraint''*: This sentence was not accurate on our part. In fact we do impose a hard constraint in the form of a quantizer with bounded rate. We apologize for this oversight and the confusion that it may have caused. We have corrected this.
>
> 13. *training method*: Indeed, $h$ is trained adversarially with a WGAN. We alternate between training $(f,g)$ and training $h$ using objective (17), where the dual form of $W_1$ given by minimizing ${\mathbb E}[{X}] - {\mathbb E}[{h(\tilde{Y})}]$ is estimated using samples.
>
> 14. *Sufficiency of separate training and relevance of theory*:  While separate training of encoders and decoders is not optimal in general and one must jointly consider the losses associated with quantization, dequantization and transport, our experimental results in Table 2 provides evidence  that in practice, the optimal rate-distortion representations at the encoder and decoder could be leveraged  without much loss of optimality. We believe that such architectural insights are interesting and demonstrate the value of our theoretical analysis.

---

### Official Review · Reviewer_mTA9 · 2021-11-03

**Correctness:** 4
**Technical Novelty And Significance:** 3
**Empirical Novelty And Significance:** 3
**Recommendation:** 6
**Confidence:** 2

**Details Of Ethics Concerns:**

No ethical concerns.

**Main Review:**

I don't see any issues with the Theorems 1-3, but I did not work through the derivation provided in the appendix. The motivation also makes sense to me since there may be benefits to treating restoration (denoising, deblurring, super-resolution, etc.) and compression jointly. In some sense this seems like the typical case, rather than a rare one, since many sensors will include some kind of signal degradation and only store / transmit a compressed representation of the captured data (e.g. the camera in a mobile phone). That said, it shouldn't be to difficult to get paired data from such a device meaning that a supervised approach can be used.

Some additional information on how shared randomness works would be helpful, though presumably this is covered in the references. I assume the sender and receiver agree on a pseudo-random number generator ahead of time and some kind of seed is transmitted, after which both sides can generate the same U.

It was not clear from the paper how the results compare to a baseline method that does *not* model restoration and compression jointly. I think this would look something like: learn to upscale low-res mnist images without a rate constraint (in an unsupervised fashion), and then separately learn to compress the high-res reconstructions.

**Summary Of The Paper:**

This paper addresses the problem of lossy compression when the source and target distributions differ. As example domains, the authors look at denoising (the distribution of noisy images and "clean" images are not the same) and super-resolution (again, the distribution of low-res and high-res images are different) both addressed as unsupervised learning problems, i.e. the noisy/clean and low/high-res training data are unpaired.

The problem is treated as optimal transport (to account for the domain shift) with an entropy bottleneck (to account for the rate constraint needed to achieve compression). The authors derive theoretical bounds for the achievable distortion in two scenarios (with and without a source of shared randomness between the sender and the receiver), and they show that a shared source of randomness strictly improves rate-distortion performance. The Wasserstein distance used in a Wasserstein GAN framework accounts for the distribution difference between the input and output domains.



**Summary Of The Review:**

The paper addresses a difficult and interesting problem, and I see no problems with the math. It's not clear to me how impactful these results are, however, which leads to my positive but low-confidence score.

---

> ### Author Response · Authors · 2021-11-18
> **Author Response -- Comments 1 and 2**
>
> 1. *Motivation and Practical Applications*: We thank the reviewer for suggesting the application of our setting involving sensor networks. To motivate the unsupervised approach, we note that images captured by the sensors  may be degraded even before compression (e.g. due to environmental conditions) so we may not be able to rely on the existence of ground truth. As one potential application, when aerial photographs are produced for remote sensing purposes, blurs are introduced by atmospheric turbulence, aberrations in the optical system and relative motion between camera and ground. It is often intractable to accurately model such degradation processes and thus unsupervised restoration provides a natural approach. Finally even in scenarios where paired training data can be collected, collection of such data can be time consuming and could require significant human effort to align or label the datasets which makes unsupervised approaches preferable.
>
> We have added a remark along these lines in Section 2 of the paper.
>
> 2. *Common randomness*: We thank you for the suggestion and have added a  remark in the paper to clarify this as suggested.

---

> ### Author Response · Authors · 2021-11-18
> **Author Response --- Comment 3 (Comparisons with baseline)**
>
> We thank the reviewer for the suggestion. We have incorporated your suggestions and included results for the baseline methods for the super-resolution task involving MNIST dataset in Appendix B5 in the revised version.  In the baseline method, we perform image restoration and compression separately. For the restoration, we build an U-Net with skip connections and train it in the unsupervised manner by adopting the Eq.(14) in the revised paper as objective i.e.,  $$L_1 = \mathbb{E}[\||X - \tilde{Y}\||^2] + \lambda_1 W_1(p_Y, p_{\tilde{Y}}).$$
>
> After the restoration network is trained to converge, we fix its weights and use it to produce restored images $\tilde{Y}$ given degraded one $X$. Afterwards, we adapt our end-to-end network as compression network by minimizing the following loss at different rates:
> $$L_2 = \mathbb{E}[\||\tilde{Y} - Y^+\||^2] + \lambda_2 W_1(p_Y, p_{Y^+}),$$
>
> where $Y^+$ is the outputs of compression network. To guarantee the distribution of reconstructed $Y^+$ is close to that of target images we have again implemented a penalty on the Wasserstein-1 distance. We experimentally selected $\lambda_1=0.05$ and $\lambda_2=0.01$ above. Once the compression network is converged, we report the final MSE distortion between $Y^+$ and $X$ using $\mathbb{E}[\||X-Y^+\||^2]$.
>  The detailed results are shown in the table below. It can be easily seen throughout  that our end-to-end system outperforms the baseline by a large margin.
>
> **Table 1**
>
>  Comparison between our end-to-end system with the baseline method. Numbers are the MSE distortion loss for a particular rate. Best results are in bold.
>
>  ***
>
>  >Super-resolution **with** Common Randomness
>
>  | Rate     | 4          | 6          | 8          | 10         | 12         | 14         | 16         | 18         |
>  | -------- | ---------- | ---------- | ---------- | ---------- | ---------- | ---------- | ---------- | ---------- |
>  | Baseline | 0.0603     | 0.0568     | 0.0544     | 0.0530     | 0.0523     | 0.0511     | 0.0503     | 0.0498     |
>  | Ours     | **0.0515** | **0.0457** | **0.0420** | **0.0394** | **0.0372** | **0.0353** | **0.0339** | **0.0342** |
>
>  | Rate     | 20         | 22         | 24         | 26         | 28         | 30         | 32         | -    |
>  | -------- | ---------- | ---------- | ---------- | ---------- | ---------- | ---------- | ---------- | ---- |
>  | Baseline | 0.0489     | 0.0485     | 0.0484     | 0.0482     | 0.0478     | 0.0476     | 0.0471     | -    |
>  | Ours     | **0.0313** | **0.0300** | **0.0297** | **0.0285** | **0.0280** | **0.0277** | **0.0269** | -    |
>
>
>
>  >Super-resolution **without** Common Randomness
>
>  | Rate     | 4          | 6          | 8          | 10         | 12         | 14         | 16         | 18         |
>  | -------- | ---------- | ---------- | ---------- | ---------- | ---------- | ---------- | ---------- | ---------- |
>  | Baseline | 0.0620     | 0.0585     | 0.0573     | 0.0555     | 0.0543     | 0.0535     | 0.0532     | 0.0520     |
>  | Ours     | **0.0558** | **0.0506** | **0.0463** | **0.0435** | **0.0415** | **0.0396** | **0.0383** | **0.0367** |
>
>  | Rate     | 20         | 22         | 24         | 26         | 28         | 30         | 32         | -    |
>  | -------- | ---------- | ---------- | ---------- | ---------- | ---------- | ---------- | ---------- | ---- |
>  | Baseline | 0.0489     | 0.0485     | 0.0484     | 0.0482     | 0.0478     | 0.0476     | 0.0471     | -    |
>  | Ours     | **0.0351** | **0.0337** | **0.0331** | **0.0328** | **0.0315** | **0.0308** | **0.0300** | -    |
>
> At this time we are still running the experiments involving the SVHN dataset. If the paper gets accepted we will include a similar comparison table in the final version.

---

> > ### Author Response · Authors · 2021-11-26
> > **Experimental results for the SVHN dataset**
> >
> > We are  reporting the results comparing our proposed scheme with the suggested baseline for the denoising application involving the SVHN dataset. Our experimental setting of denoising baseline shares a similar architecture as the super-resolution experiment discussed above. The loss functions and training strategies can be found in our last response. We implemented a new restoration network , which is also a U-net but contains more convolutional layers and has about 3.70 million parameters (as opposed to 0.14 million parameters for the previous experiment). Second, we experimentally selected $\lambda_1=0.03$ and $\lambda_2=0.005$ in the loss functions $L_1$ and $L_2$ stated in the above response. The detailed results are shown in the table below. It can be observed that our method again outperforms the baseline.
> >
> >  **Table 2: Image Denoising Experiment**
> >
> >  Comparison between our end-to-end system with the baseline method. Numbers are the MSE distortion loss for a particular rate. Best results are in bold.
> >
> >  ***
> >
> >  >Image Denoising **with** Common Randomness
> >
> >  | Rate     | 12          | 18          | 24          | 30         | 36         | 42         | 48         | 54         |60         |
> >  | -------- | ---------- | ---------- | ---------- | ---------- | ---------- | ---------- | ---------- | ---------- | ---------- |
> >  | Baseline | 0.0242     | 0.0213     | 0.0189     | 0.0173     | 0.0162     | 0.0154     | 0.0148     | 0.0142    |0.0136    |
> >  | Ours     | **0.0219** | **0.0195** | **0.0175** | **0.0163** | **0.0154** | **0.0147** | **0.0138** | **0.0135** | **0.0129** |
> >
> >  | Rate     | 66         | 72         | 78         | 84         | 90         | 96         | 102         | 108    | 114    |
> >  | -------- | ---------- | ---------- | ---------- | ---------- | ---------- | ---------- | ---------- | ---- | ---- |
> >  | Baseline | 0.0134     | 0.0130     | 0.0127     | 0.0124     | 0.0120     | 0.0118     | 0.0116     | 0.0111    | 0.0110    |
> >  | Ours     | **0.0126** | **0.0123** | **0.0118** | **0.0117** | **0.0115** | **0.0112** | **0.0109** | **0.0107**    | **0.0106**    |
> >
> >
> >
> >  >Image Denoising **without** Common Randomness
> >
> >  | Rate     | 12          | 18          | 24          | 30         | 36         | 42         | 48         | 54         |60         |
> >  | -------- | ---------- | ---------- | ---------- | ---------- | ---------- | ---------- | ---------- | ---------- | ---------- |
> >  | Baseline | 0.0252     | 0.0216     | 0.0202     | 0.0185     | 0.0178     | 0.0164     | 0.0158     | 0.0154     | 0.0149     |
> >  | Ours     | **0.0230** | **0.0208** | **0.0189** | **0.0175** | **0.0165** | **0.0157** | **0.0151** | **0.0145** | **0.0140** |
> >
> >  | Rate     | 66         | 72         | 78         | 84         | 90         | 96         | 102         | 108    | 114    |
> >  | -------- | ---------- | ---------- | ---------- | ---------- | ---------- | ---------- | ---------- | ---- | ---- |
> >  | Baseline | 0.0147     | 0.0144     | 0.0140     | 0.0135     | 0.0133     | 0.0129     | 0.0126     | 0.0120    | 0.0117    |
> >  | Ours     | **0.0137** | **0.0133** | **0.0130** | **0.0127** | **0.0123** | **0.0120** | **0.0118** | **0.0116**    | **0.0114**    |

---

### Official Review · Reviewer_SKN1 · 2021-11-03

**Correctness:** 4
**Technical Novelty And Significance:** 3
**Empirical Novelty And Significance:** 3
**Recommendation:** 6
**Confidence:** 3

**Main Review:**

The theorems shown in this work look interesting. For example, Theorem 1 indicates that, without common randomness, the optimal solution can be decomposed with compression (quantization and dequantization) and transport. Authors also demonstrate that common randomness is helpful at reducing the rate for a given distortion. Experimental results can corroborate the theorems.

I am just a bit confused by the term "cross domain". I was thought it means different data sources with distinct spatial structure and appearance. But it seems authors refer to distributional shift due to noise and blurring.

My only suggestion is that, it seems Wang et al. (2021b) also model the shift in distribution due to degradation as an optimal transport problem. It would be much better if authors can specify more differences with respect to this work.

Possible typos:
Page 6, second last line, "f be an encoder, Q, and g be a decoder", do you want to mention "Q be a quantizer"?

**Summary Of The Paper:**

This work considers compression and restoration of degraded input in a joint framework, and formulated this problem as an oprimal transport with a constraint on the rate. Authors systematically analyzed the scenarios with and without common randomness at the encoder and decoder, and provide a few new insights both theoretically and empirically.

**Summary Of The Review:**

The theorems mentioend in this work is insightful. Authors also discussed how to implement compression and restoration jointly by a deep neural network.

---

> ### Author Response · Authors · 2021-11-18
> **Response by Authors**
>
>
> 1. *Suggestion regarding  the term: cross-domain*: We thank the reviewer for alerting us of possible confusion  in using this term.  Our original thinking  was that the theoretical framework in Section 2 considers a general setting (where $p_X$ refers to the distribution of the input domain and $p_Y$ refers to the distribution of the reconstruction domain), which could capture a variety of cross-domain applications. Nevertheless we agree that the focus of this paper (both in experiments and motivation) is mainly on image restoration tasks involving a distribution shift. Based on your suggestion we have decided to revise the title as: *Lossy Compression with Distribution Shift as Entropy Constrained Optimal Transport*. We hope that the revised title addresses your concern. We have noted in the Conclusions section of the paper that although our focus has been on image restoration tasks, our theory could be applicable to a broader set of cross-domain tasks.
>
> 2. *Differences from Wang et al. (2021b)*: The work of Wang et al (2021b) relates image restoration to optimal transport and does not consider compression. In contrast, our work considers the setting where both compression and restoration must be performed simultaneously and views it as  entropy constrained optimal transport. Accordingly our results are very different. Our Theorem 1  demonstrates the extent to which the compression and transport problems can be decoupled without loss of optimality, while Theorem 3 builds upon the ideas in the compression literature (e.g., Theis and Agustsson (2021)) to establish the utility of common randomness in this setting. We believe that such results are not present in Wang et al (2021b) as they do not consider compression. Our experiments are designed to study achievable rate-distortion tradeoffs and different from Wang et al (2021b).
>
> 3. Thanks for the correction; Q should be a quantizer.

---

### Official Review · Reviewer_YvyS · 2021-11-07

**Correctness:** 4
**Technical Novelty And Significance:** 3
**Empirical Novelty And Significance:** 3
**Recommendation:** 8
**Confidence:** 3

**Main Review:**

(+) pros / (-) cons
-------------------
(+) well written and developed paper

(+) topic interesting and important, addressed innovatively

Further questions for clarification/discussion
-----------------------------------------------
Binary example is hepful but the use case is for more complex (continuous) X and Y distributions. Would it be possible to find an illustrative example in the continuous space?

Minor text problems / typos
---------------------------
Page 1, 7 lines from bottom: application -> applications

Page 6, 2 lines from bottom: Q -> Q is a quantizer

**Summary Of The Paper:**

The paper explores the problem of learned compression scheme for the specific setup where the source and target distributions are not the same (such as in restoring a degraded image). The authors formulate the problem as an optimal transport problem with entropy constraints to control the compression rate. They further argue that the use of common randomness in the encoder and decoder improves the rate-distortion tradeoff and dcoument it on denoising and super-resolution experiments.

**Summary Of The Review:**

I find the paper contributes innovatively to the field nds is worth the attention of the community. Hence I recommnend it for acceptance for the conference.

---

> ### Author Response · Authors · 2021-11-18
> **Additional Numerical Examples**
>
>
> We have added additional numerical examples in the paper involving continuous valued sources as noted below:
>
> 1. In Appendix  A6 we have considered the case when the variables $X$ and $Y$ are sampled from continuous valued uniform density functions. We make use of the decomposition in Theorem 1 to establish an upper bound on the minimum achievable distortion $D_\mathrm{ncr}$ and study its properties.
> 2. For Gaussian distributions in the asymptotic optimal transport setting (see Appendix A.7 for relevant definitions and results) we present results that are qualitatively similar to the binary case in Appendix A.8.
>
> In the main text, we have updated Section 2.2 to provide more intuition on the results for the binary case and also noted Appendices A6, A7 and A8 mentioned above.

---

### Author Response · Authors · 2021-11-18
**Summary of Major Changes**


We thank the reviewers for their constructive comments that has helped us prepare a revised version of the paper. The major changes are as follows.

1) Based on a comment by reviewer SKN1, we have revised the title of the paper to *Lossy Compression with Distribution Shift as Entropy Constrained Optimal Transport* to better reflect the motivation and experimental sections of the paper.
2)  Based on a comment by reviewer YvyS, we have added Appendices A6, A7 and A8 to include additional numerical examples involving continuous valued sources.
3) Based on the comment by reviewer mTA9, we have added comparisons with the suggested baseline scheme in our experiments in Appendix B5. We have presented the results for the super-resolution tasks involving MNIST dataset. ~We are continuing to run the experiments for the denoising task using the SVHN dataset and will include them if the paper gets accepted.~ We have also included the similar results for the denoising application with the SVHN dataset in our response to the reviewer.
4) Based on the suggestion of reviewer GKce we have rewritten Section 2.2 to focus on the intuition and takeaways for our example involving binary sources and presented the technical results in the Appendix. We have also addressed several other clarifications suggested by the reviewer.
5) Based on the comments of reviewer LfvG  we have added a clarification on the assumptions involving common randomness in our setup.

In addition, we have addressed all the comments in detail in the revised version of the paper. The major changes in the paper are highlighted using the blue font. We separately answer each of the comments by the reviewers below.

---

### Decision · Program_Chairs · 2022-01-20

**Decision:**

Accept (Poster)

**Comment:**

This paper discusses the problem of cross-domain lossy compression on the basis of its reformulation as an entropy-constrained optimal transport. Two average distortion measures (without and with common randomness) are defined (Definitions 2 and 3), and some of their properties are investigated, as summarized in Theorems 1-3. The authors also demonstrated in Section 2.2 that in the Bernoulli-Hamming case the common randomness can indeed improve the performance under some conditions. Results of the numerical experiments on super-resolution and denoising are presented to illustrate the principles derived from the theoretical considerations.

This paper received 5 reviews, with score/confidence being 8/3, 6/3, 6/2, 8/3, 3/4, which exhibit a relatively large spread across the borderline. Upon reading the reviews and the author responses, as well as the paper itself, I think that this paper proposes an interesting framework of optimal transport with entropy bottleneck, as well as architectural designs supported by the theoretical development, with potential image-processing applications. The authors have provided further numerical results in their response.

My main concern is that the arguments in this paper are somehow confusing in that they borrow several notions and terms from the context of lossy compression and rate-distortion theory in the field of information theory, and use them in quite different meanings without explicitly stating so. (It seems to me that this would have been one major reason for the negative evaluation by Reviewer LfvG.) Examples are:
1. **Target distribution:** In rate-distortion theory the target distribution $p_Y$ is not fixed, whereas it is fixed in this paper.
2. **Rate constraint:** In rate-distortion theory the rate constraint is imposed in terms of the mutual information $I(X;Y)$ between the source $X$ and the target $Y$, in a form $I(X;Y)\le R$. The justification of this particular form of the rate constraint rather than any other forms is that it is compatible with the operational achievability/converse arguments via explicit construction of encoder/decoder pairs. In this paper, on the other hand, the authors consider a Markov chain $X\to Z\to Y$ and impose the rate constraint on the entropy $H(Z)$ of the intermediate random variable $Z$. Under the Markov assumption one has $I(X;Y)\le H(Z)$, so that the rate constraint in this paper is stronger than that in rate-distortion theory, and one would have no control over how tight/loose the adopted constraint $H(Z)\le R$ is against $I(X;Y)\le R$. In relation to this, the expression "identify the tradeoff between the compression rate and minimum achievable distortion" (page 2, line 12) would be at best misleading, as the arguments in this paper might be suboptimal, not necessarily providing the theoretically best achievable results.
3. **Extension versus single-shot:** In rate-distortion theory one usually considers $n$th extension of a source and a block encoder/decoder pair with blocklength $n$. On the other hand, this paper considers what is called the "single-shot" setting, in which one does not consider extension of sources. There are some pieces of work on lossy compression in the single-shot setting [C1][C2], so that I would be interested in how such pieces of work and the development in this paper will be related, an issue not explored at all in this paper.

As a result, although the quantities $D_{\mathrm{ncr}}$ and $D_{\mathrm{cr}}$ defined in Definitions 2 and 3 would look quite like the distortion-rate functions in rate-distortion theory, they are actually not the distortion-rate functions at all. Although the authors, perhaps carefully, did not call them distortion-rate functions, there should still be some explicit explanation on the difference of their framework from the standard one in information theory.

[C1] Nir Elkayam and Meir Feder, "One shot approach to lossy source coding under average distortion constraints," IEEE International Symposium on Information Theory, 2389-2393, 2020. [link](https://ieeexplore.ieee.org/stamp/stamp.jsp?tp=&arnumber=9173943)

[C2] ibid., "One shot approach to lossy source coding under average distortion constraints," [Online]. Available: https://arxiv.org/abs/2001.03983

Another concern of mine, from the viewpoint of the theory of optimal transport, is regarding the optimal transport map. It is known that under fairly general conditions the optimal transport *plan* exists. However the optimal transport *map* is not guaranteed to exist, and even if it exists it can be highly irregular. (See, e.g., Villani, 2009) The descriptions in this paper, such as "computing the optimal transport map is not straightforward" and "learn approximately optimal mappings" on page 4 would be too naive in that they would assume existence and approximability of the optimal mapping.

Despite these concerns, most reviewers agree that this paper presents an interesting piece of work. I would therefore like to recommend acceptance of this paper, and would appreciate it if the authors consider addressing appropriately the above concerns of mine in the final version.

---

> ### Public Comment · ~Ashish_J_Khisti1 · 2022-03-10
> **Clarifications**
>
> We thank the area chair for bringing up some issues that have caused confusion during the review process and deciding to accept our paper. We would like to kindly clarify the issues raised by the AC.
>
> **Point 1**: The first major point that the AC brings up is that in our problem formulations (Definition 2 and Definition 3) we did not impose a rate-constraint of the form $I(X;Y)\le R$, but instead assumed a "stronger" constraint of the form $H(Z) \le R$, (where $Z$ is the intermediate representation of $X$ satisfying $X \rightarrow Z \rightarrow Y$.) The AC states that while  our condition is sufficient, it does not provide any guarantees on how tight the constraint $I(X;Y) \le R$ is. In that sense some of our statements were considered to be correct but potentially misleading to the readers.
>
> **Response**: We believe that when the AC is referring to a constraint on $I(X;Y)$ in rate-distortion theory he/she has in mind that classical asymptotic setting where the encoder observes $n$ i.i.d samples of the source (with $n\rightarrow\infty$). We in fact do cover this setting in our submission. In particular in Theorem 7 in the Appendix we precisely demonstrate how our Problem setup In Definition 3 (i.e. with common randomness), when applied to the asymptotic setting leads to a constraint that $I(X;Y)\le R$ as noted by the AC. Likewise Theorem 5 in the Appendix gives counterpart results for the case without common randomness. We have provided numerical evaluation of these for quadratic-Gaussian source in Theorem 8 in the Appendix.  Please note that Definition 2 and Definition 3 and the associated results illustrated in Figure 2 consider the problem setup in the general case. We believe that our problem formulation appears naturally and recovers results of (asymptotic) rate-distortion theory as explained. We hope this addresses any confusion that the AC may have on why a constraint on mutual information was not imposed in Definition 2 and 3.
>
>
> **Point 2** Connection to one-shot methods in Information Theory and specifically references [C1] and [C2]
>
> **Response**: References [C1] and [C2] noted above consider a rate-constrained lossy source coding setup where a source sample $X$ must be compressed using a fixed length code of rate $R$ bits to minimize the expected distortion between the source and the reconstruction. They provide an expression for the expected distortion over the ensemble of random codebooks consisting of $e^R$ reconstruction points. Our setup is different in that we use a variable length compression scheme and bound the entropy of the representation variable. Furthermore at-least in the setup without common randomness at the encoder and the decoder, it seems that the construction in [C1] and [C2] would restrict the reconstruction output to be a discrete variable over finite support. It may not satisfy the given target distribution. Due to the difference in setup, we note that our results are not comparable with references [C1] and [C2].
>
> The most relevant reference in  to our understanding is the paper by Li and El Gamal (2018), where a variable-length lossy compression setting is considered. We have already cited this in the paper after Theorem 3. However that work does not impose a constraint on the output distribution. More importantly the proof technique of Li and ElGamal uses the strong function representation lemma (SFRL), which requires common randomness at the encoder and decoder and provides upper bounds on the achievable rate. We use very different analysis techniques to develop exact results for both the settings with and without common randomness. Such analysis was necessary as we believe that a straightforward application of SFRL may not give useful insights in our setting, particularly in the example involving binary sources.
>
> **Point 3** Concern about existence of optimal transport map
>
> **Response** The above comments on page 4 were made following Theorem 2 where a squared-euclidean distance measure is assumed. The following paper (See Appendix A2) notes that optimal-transport map exist in such as setting.
>
> Freirich, Dror, Tomer Michaeli, and Ron Meir. "A theory of the distortion-perception tradeoff in wasserstein space." Advances in Neural Information Processing Systems 34 (2021).
>
> We will add a suitable clarification in the paper to note that the comments pertain to the results stated in Theorem 2.